# Soil Erosion Characteristics and Scenario Analysis in the Yellow River Basin Based on PLUS and RUSLE Models

**DOI:** 10.3390/ijerph20021222

**Published:** 2023-01-10

**Authors:** Yanyan Li, Jinbing Zhang, Hui Zhu, Zhimin Zhou, Shan Jiang, Shuangyan He, Ying Zhang, Yicheng Huang, Mengfan Li, Guangrui Xing, Guanghui Li

**Affiliations:** 1College of Geography and Environmental Science, Henan University, Kaifeng 475004, China; 2Key Laboratory of Geospatial Technology for the Middle and Lower Yellow River Regions, Ministry of Education, Henan University, Kaifeng 475004, China; 3Regional Planning and Development Center, Henan University, Kaifeng 475004, China

**Keywords:** PLUS model, RUSLE model, land use, soil erosion, prediction, YRB

## Abstract

Soil erosion is an important global environmental issue that severely affects regional ecological environment and socio-economic development. The Yellow River (YR) is China’s second largest river and the fifth largest one worldwide. Its watershed is key to China’s economic growth and environmental security. In this study, six impact factors, including rainfall erosivity (R), soil erosivity (K), slope length (L), slope steepness (S), cover management (C), and protective measures (P), were used. Based on the revised universal soil loss equation (RUSLE) model, and combined with a geographic information system (GIS), the temporal and spatial distribution of soil erosion (SE) in the YR from 2000 to 2020 was estimated. The patch-generating land use simulation (PLUS) model was used to simulate the land-use and land-cover change (LUCC) under two scenarios (natural development and ecological protection) in 2040; the RUSLE factor P was found to be associated with LUCC in 2040, and soil erosion in the Yellow River Basin (YRB) in 2040 under the two scenarios were predicted and evaluated. This method has great advantages in land-use simulation, but soil erosion is greatly affected by rainfall and slope, and it only focuses on the link between land-usage alteration and SE. Therefore, this method has certain limitations in assessing soil erosion by simulating and predicting land-use change. We found that there is generally slight soil erosivity in the YRB, with the most serious soil erosion occurring in 2000. Areas with serious SE are predominantly situated in the upper reaches (URs), followed by the middle reaches (MRs), and soil erosion is less severe in the lower reaches. Soil erosion in the YRB decreased 11.92% from 2000 to 2020; thus, soil erosion has gradually reduced in this area over time. Based on the GIS statistics, land-use change strongly influences SE, while an increase in woodland area has an important positive effect in reducing soil erosion. By predicting land-use changes in 2040, compared to the natural development scenario, woodland and grassland under the ecological protection scenario can be increased by 1978 km^2^ and 2407 km^2^, respectively. Soil erosion can be decreased by 6.24%, indicating the implementation of woodland and grassland protection will help reduce soil erosion. Policies such as forest protection and grassland restoration should be further developed and implemented on the MRs and URs of the YR. Our research results possess important trend-setting significance for soil erosion control protocols and ecological environmental protection in other large river basins worldwide.

## 1. Introduction

Soil erosion (SE) severely affects the normal development of agriculture and industry, decreases soil fertility, leads to ecological degradation, and damages water and transportation engineering facilities. It is a global issue that has led to land degradation and impairment of ecosystem services and posed serious threat to human public health safety [1,2]. The total amount of SE resulting from land-usage alterations has increased by 2.5% from 2001 to 2012 [3]. China is a nation featuring the world’s most severe SE. SE can result in extensive ecological limitations and regional socio-economic development can be severely constrained [4]. The key to soil erosion control and soil conservation macroscopic decision making is predominantly quantitative examination of soil erosion and analysis of soil erosion spatial distribution [5,6]. Conducting soil erosion research is significant to understand the type, mechanism, spatiotemporal distribution, and dynamic evolution process of erosion occurrence in a region. Research can play a role in identifying erosion-prone areas, and to formulate soil and water conservation planning, restoration, and reconstruction measures.

Land-use change represents a link between human activities and ecological processes, and may affect the formation and function of ecosystems, which affects ecosystem services [7,8]. As an important factor affecting soil erosion, structural change in land use can influence soil erosion intensity [9]. At the slope scale, different spatial combinations of vegetation patterns can affect slope erosion sand production [10]. At the watershed scale, there is a correlation between land patch fragmentation and soil erosion intensity [11]. At the regional scale, scenario simulation results have exhibited that converting grassland to cropland per km^2^ can lead to a rise of 0.38 t·km^−2^·a^−1^ in mean erosion intensity [12]. Effective control of SE, more sustainable conservation practices, sustainable use of soil and water resources, and maintaining ecosystem health are key environmental issues facing countries worldwide. However, with increasing population–resource–environment pressure, the African continent, and especially the upper Nile hilly region, has become a key area of international concern for ecosystem health and food security assurance [13]. Driven by human activity, changes in land-use types, surface microtopography, construction of drinking water storage projects, soil structure, and surface vegetation are directly related to surface soil loss. The main cause of soil loss in sub-Saharan Africa basin is land-use change caused by human activities [14]. Owing to economic, technological, and demographic factors, Southeast Asian countries are experiencing extensive land-use changes that accelerate the extent of soil erosion hazards [15]. Borrelli stated that global soil erosion increased by 20% from 2001 to 2012, with approximately 38% of agricultural land being degraded due to soil erosion [3]. The annual soil replacement cost of soil erosion in Europe is likely to reach EUR 20 billion [16], and the SE spatial area in China is 294.9 × 10^4^ km^2^ [17], which has become a severe threat to the ecological security of the country and sustainable societal development.

Many research methods have been employed for evaluating SE, for example, the universal erosion formula (USLE) [18], the RUSLE model [19], the European soil erosion model [20], and the Chinese soil erosion equation [21]. The most widely used of these is the USLE/RUSLE model. The RUSLE model is obtained through continuous modification of the USLE model and has been predominantly applied to the hydraulic erosion of stable gully systems, such as sloping arable land, woodland, and bare ground on hillslopes. Researchers have applied the RUSLE model widely in SE studies in several watersheds. Onori et al. adopted the RUSLE model for assessing potential annual SE in the Comunelli catchment and tested the reliability of the method [22]. Humphrey et al. conducted a spatial–temporal assessment of monthly SE risk modeling in the Wenham Bay watershed, Kenya, based on the RUSLE model [23]. Feng et al. conducted a soil erosion simulation by utilizing the RUSLE model and the 137Cs tracer approach in a karst watershed in Guangxi [24].

Research has been conducted nationally and internationally on how human activities affect SE, predominantly focusing on land-use change and how water and soil conservation steps affect SE. This has undergone a development process from qualitative to quantitative and from single description to detailed evaluation. There is a substantial influence of spatial patterns of land usage on SE. Previous studies using SE equations to estimate soil erosion intensity mainly used land use as a key parameter in analyzing the evolution of regional soil erosion intensity [25]. However, few researchers have explored how land-use alteration affects SE by studying the links between land-usage shifts and SE change or the change in SE intensity index based on different land-usage forms.

The Yellow River (YR) is also a river with the highest sand content, with sand production accounting for 6% of the total sand production in river systems worldwide [26]. A major ecological problem is SE in the Yellow River Basin (YRB). To alleviate problems from SE, the government has implemented large-scale projects to return farmland to grassland, for example, restore forest and grassland ecosystems and transform degraded farmland using soil and water conservation measures [27]. Implementing the program to return farmland to grassland promotes land-usage changes in the YRB [28]. Whether land-usage change reduces regional soil erosion and brings into play the ecological effect of soil conservation is an important element to consider during an evaluation of the ecological effect at the later stages of a RULSE project. Soil erosion in the YRB is mainly assessed according to the RULSE model or through analyzing the impact of land-usage alterations on SE. However, there are fewer studies on simulated and predicted land-usage alterations in line with the PLUS model to predict future soil erosion in the YRB.

In this investigation, the spatiotemporal alterations in soil erosion in 73 prefecture-level cities in the YRB were analyzed from 2000 to 2020 using the RUSLE model to explore the links between land-usage change and SE (Figure 1). Our specific objectives include the following: (1) to reveal the changes in soil erosion in the YRB from 2000 to 2020, and the characteristics of changes under different scenarios; (2) to discuss the relationship between land-use change and soil erosion, and to analyze the impact of land-use change on soil erosion quantitatively; (3) based on the PLUS model, land-use change is predicted, and soil erosion in the YRB in 2040 is predicted in combination with the *P*-factor of the RUSLE model. It reveals the changes in SE under different scenarios and supplies a scientific basis for more science-driven erosion control, ecological zoning, and coordinated improvement of the economic environment in the YRB. The findings can also aid in deepening our understanding in formulating relevant policies for effective management in areas with severe soil erosion.

## 2. Materials and Methods

### 2.1. Study Area

The YR has a total length of 5464 km [29]. On the upper reaches of the YR, there are many mountain ranges, undulating terrain, and severe vegetation degradation. On the middle reaches, the geological structure is fragmented, and the soil texture is relatively loose, making it the most susceptible to SE. On the lower reaches of the YR, the alluvial plain has substantial sediment accumulation, with the riverbed being constantly elevated, which is referred to as the world-famous “hanging river on the ground.” Given that the Yellow River has been diverted many times, this has a profound influence on the life and production of coastal cities in the area. We combined study feasibility with the integrity of the administrative unit area and the direct relationship between inter-regional economy and the Yellow River based on the natural watershed classification [30]. We used 73 municipal districts, including prefecture-level cities, states, and leagues affected by the Yellow River, as the study area (30°35′–43°28′, 89°35′–121°00′), encompassing a study area of 203.36 × 10^4^ km^2^ (Figure 2). The Yellow River is the sandiest river worldwide, and along with the destruction of vegetation and steep slopes, the YRB is a most active area of SE in China [31].

### 2.2. Data Sources

This study mainly uses vector data, DEM data, land use data, meteorological data, NDVI data, soil data, road network data, and socio-economic data. Among them, vector data refer to the boundary contour of the study area and the location of China’s boundary contour. According to the “China Land Use/Land Cover Remote Sensing Monitoring Data Classification System” the land use data were reclassified into six categories: cropland, forestland, grassland, water area, built-up land, and unused land. The soil data include the extraction of clay, silt, sand, and organic carbon. The meteorological data refer to the temperature and precipitation data in the study area, which were obtained by interpolation using the Anusplin software according to the meteorological station data. The road network data mainly refer to high-speed railway, main trunk roads, and secondary roads. The datasets were obtained online with a resolution of 1 × 1 km, and for maintaining data resolution consistency, data higher than 1 km resolution were uniformly resampled using the ArcGIS 10.2 software. The data was collected for 2000–2020, and the specific data sources are exhibited in Table 1.

### 2.3. RUSLE Model

The RUSLE model is an internationally popular model [19,32] that compensates for suiting multiscale simulation investigations but it is also constrained by field observation in large-scale applications [33,34]. It can achieve better results in simulations at different scales, and considers precipitation erosion forces, soil erodibility, slope length, soil and water conservation steps, soil conservation measures, and other main factors affecting soil erosion [35,36]. The formula is as follows:(1)A=R×K×L×S×C×P
in which *A* indicates the annual mean SE in t hm^−2^ a^−1^; *R* indicates the rainfall erosivity factor in t ha MJ^−1^ mm^−1^; *K* indicates the soil erodibility factor in t ha MJ^−1^ mm; *L* indicates the slope length factor (dimensionless); *S* indicates the slope steepness factor (dimensionless); *C* indicates the land cover and management factor (dimensionless); and *P* indicates the conservation practice factor (dimensionless).

#### 2.3.1. Rainfall Erosivity Factor (*R*)

Rainfall erosion force is a driver of SE and directly affects the SE modulus magnitude [37]. Calculation of the rainfall erosion factor (*R*) can be divided into two categories, namely the classical calculation method of EI30 and the simple algorithm of conventional meteorological data. As it is difficult to obtain 30 min rainfall intensity I30 and rainfall kinetic energy E information, many researchers have established simple models based on conventional rainfall information according to regional rainfall erosion characteristics. We used the Wischmeier and Smith (1978) equation [19,38], which expressions are as follows:(2)$$R=\sum_{i=1}^{12}1.735\times 10^{1.5\mathrm{lg}\left(\frac{p_{i}{}^{2}}{p}\right)-0.8188}$$
in which *p_i_* represents the monthly precipitation (mm) and *p* denotes the annual one (mm). Multiply this unit by a coefficient of 17.02 converts it to the international unit MJ-mm/(hm^2^·h·a)

#### 2.3.2. Soil Erodibility Factor (*K*)

Soil erodibility is influenced by soil properties, which are determined by soil type. Soil structure is classified into clay, very fine sand, and silt [39]. Silt and very fine sand are defined by the United States as 0.002–0.05 and 0.05–2.0 mm, respectively. Clay is referred to as particles with a size of <0.002 mm [40].

To avoid errors associated with using supervised classification methods, such as maximum likelihood, to obtain land use data, this study can estimate *K* values more easily based on the Erosion Productivity Effect Calculator model (EPIC model) [41], as calculated below:(3)$$K=\left\{0.2+0.3*\exp\left[-0.0256*SAN*(1.0-\frac{SIL}{100})\right]\right\}\times {\left(\frac{SIL}{CLA+SIL}\right)}^{0.3}\times \left\{1.0-\frac{0.25*C}{C+\exp(3.72-2.95*C)}\right\}\times \left(1.0-\frac{0.7*S_{n}}{S_{n}+\exp(22.9S_{n}-5.51)}\right)\times 0.1317$$
where CLA, SIL, and SAN represent concentrations (%) of clay, silt, and very fine sand in the soil, respectively, and Sn = 1 − SAN/100 and *C* represent the soil organic carbon concentration (%). Since this formula was proposed by the United States for particle size and sand size, multiplying it by 0.1317 converts the American units into international units (t ha h ha^−1^ MJ^−1^ mm^−1^). Soil organic matter contains a high amount of organic carbon, and its content (*C* value) can be obtained by dividing the content of organic matter by 1.724.

#### 2.3.3. Slope Length (*L*) and Steepness (*S*) Factors

Regional topography can affect erosion caused by rainfall. Slope runoff is a driver of sediment transport and SE [42]. Slope length determines the variation in flow energy and the transport mechanics of flow and sediment. Slope length and steepness factors are important topographic parameters for estimating soil erosion in the RUSLE model. *L*-values are computed utilizing the classical equation presented by Wischmeier et al. [43], which expressions are as follows:(4)$$L={\left(\frac{\lambda }{22.13}\right)}^{\alpha }$$
(5)$$\alpha =\frac{\beta }{\beta +1}$$
(6)$$\beta =\left(\sin\theta ÷0.0896\right)/\left[3.0\times {\left(\sin\theta \right)}^{0.8}+0.56\right]$$
where *λ* represents the slope length in m; flow accumulation indicates the cumulative uphill area of a unit; cellsize is the spatial resolution of the grid; *α* indicates the slope length index; *β* indicates the ratio of rill to inter-rill erosion; *θ* indicates the slope value extracted from DEM in °; and 22.13 m indicates the standard slope length [44].

S is computed by applying the slope equation of the RUSLE model. The slope factor is computed by employing the equation proposed by McCool et al. for areas below 10° [45], and the slope factor for areas above 10° is computed by adopting the equation improved by Liu et al. based on the steep slopes of the Loess Plateau [22]. The calculation equation is as follows:(7)$$S=\left\{\begin{array}{ll} 0.03+10.8\times \sin\theta & \theta <5^{{}^{\circ}}\\ -0.50+16.8\times \sin\theta & 5^{{}^{\circ}}\leq \theta <10^{{}^{\circ}}\\ -0.96+21.9\times \sin\theta & \theta \geq 10^{{}^{\circ}} \end{array}\right.$$
where *θ* represents the slope value (°).

#### 2.3.4. Cover Management Factor (*C*)

Erosion suppression degree is influenced by vegetation cover or field management measures, and the magnitude of the *C* value is influenced by effects such as vegetation growth, vegetation type, and cover. NDVI is closely related to the surface vegetation cover and can reflect the cover pattern as well as the vegetation type and growth within a unit image element [46,47]. Therefore, NDVI is used for C-factor estimation [48,49], with the following equation:(8)$$C=\exp\left(-2.5\times \frac{NDVI}{1-NDVI}\right)$$
in which *NDVI* represents the normalized vegetation index, with its range being 0–1.

#### 2.3.5. Conservation Measure Factor (*P*)

*P* is used to explain the inhibitory effects of soil and water preservation steps on SE. There is currently no universal standard for P-factor assignment in large-scale studies. To reflect the distribution pattern of soil and water preservation steps, the *P*-factor value of woodland, grassland, and unused land is 1 and that of watershed and construction land is 0 [50]. Given that arable land accounts for a relatively high land area, the *P*-values for arable land with different slopes were calculated based on the empirical formula proposed by Lufafa [51], which is as follows:(9)P=0.03×S+0.2
where *S* is the slope (%).

### 2.4. Land-Use Change Matrix (LUCM)

We used the LUCM to study the transfer in and out of each land type from 2000 to 2020 [52]. The transfer matrix is expressed as follows:(10)$$S_{ij}=\left(\begin{array}{cccc} S_{11} & S_{12} & \cdots & S_{1n}\\ S_{21} & S_{22} & \cdots & S_{2n}\\ \vdots & \vdots & \vdots & \vdots \\ S_{n1} & S_{n2} & \cdots & S_{nn} \end{array}\right)$$
in which *S* indicates the land acreage (km^2^); *n* indicates the total quantity of land-usage types after and before the transfer; and *j* and *i* indicate the land-use types after and before the transfer, respectively.

### 2.5. PLUS Model

Previous prediction models have played a limited role in exploring the causes of land-use change, and it is difficult to dynamically simulate patch-level changes of multiple land-use types in time and space, especially for forestland, grassland, and other natural land-use types. The PLUS mode is a new model constructed in line with meta-cellular automata [53]. By analyzing the spatial characteristics of land-usage expansion parts and the drivers between the two phases of land use data, the random forest algorithm is employed for sampling and calculating land-usage expansion to obtain the development probabilities of various types of land usage one by one. Then, the comprehensive probability of land-usage alteration is obtained in line with adaptive inertial competition mechanisms of roulette. Finally, the random patch generation, the transition matrix, and the threshold are combined. The decrement mechanism realizes optimization and determines the final land-use method. Compared to the simulations of various models commonly used in current related research, the PLUS mode features a higher simulation accuracy [54]. Its simulation results can better support related studies on future land-use changes. The main advantages of this model are as follows: (1) A new analytical strategy can be used to better explore the incentives of various land-use changes. (2) It includes a new multi-category seed growth mechanism, which can better simulate the patch-level changes of multi-category land use. (3) Coupled with the multi-objective optimization algorithm, the simulation results can better support planning policy to achieve sustainable development [55].

The PLUS model prediction scenario settings and processes are as follows:(1)We extract the land use expansion from 2000 to 2020 according to the land expansion module in the PLUS software. Then, combined with the land expansion analytical strategy module, 12 driving factors are selected, including elevation, slope, aspect, annual average precipitation, annual average temperature, GDP, population, distance from expressway, distance from railway, distance from main roads, and distance from water area and soil type, to generate the development probability of each type of land.(2)Based on the CA module which is based on multi-class random patch seeds in the PLUS software, the land development probability obtained in the previous step is taken as the basic condition, and then the water area is set as the restricted development condition to obtain the diffusion coefficient. Finally, the domain weight is calculated according to the proportion of the expansion area of each land type. At the same time, two change scenarios of natural development and ecological protection are established.

Scenario of natural development. It implies the trend and characteristics of land-use change with no significant change in the factors affecting land-use change during 2000–2020. The land-use structure by 2040 is predicted according to the transfer probability from 2000 to 2020.

Scenario of ecological protection. According to the requirements of ecological protection, the transfer of forestland, grassland, and water areas would be strictly controlled. Although grassland can be converted into forestland and forestland can be converted into grassland, it cannot be converted into other land types.

(3)We select the year 2000 as the base start time and use the development probability of each land use from 2000 to 2020 to predict the land use in 2020. The verification module in the PLUS software is used to input the actual land use data and predict land use data in 2020, and its results are verified by Kappa coefficient to evaluate the simulation results. On this basis, we further simulate land use under the natural development scenario and the ecological protection scenario in 2040. The simulation results are used to calculate the P factor of RUSLE, because this factor is greatly affected by human activities. By predicting land use, and then predicting soil erosion under the influence of human activities, the relationship between land-use change and soil erosion is discussed.

## 3. Results

### 3.1. Analysis of SE Temporal and Spatial Characteristics

By calculating the soil erosion in the YRB from 2000 to 2020 and calculating the specific amount of soil erosion, the results of SE changes in the study area are shown in Table 2. The actual SE intensity in the YRB was graded, and the SE results were tallied based on the standard for SE gradation and classification (SL 190-2007) released by the Ministry of Water Resources, China (Table 3). The SE intensity can be classified into six grades, where A < 5 is very slight; 5 ≤ A < 25 is slight; 25 ≤ A < 50 is moderate; 50 ≤ A < 80 is strong; 80 ≤ A < 150 is very strong; and A ≥ 150 is severe.

According to Table 2, the soil erosion in 2000, 2010, and 2020 is 5.28, 4.8, and 4.65 t hm^−2^ a^−1^, respectively, all of which represent low levels of erosion. The total soil erosion is 10.13 × 10^8^ t, 9.20 × 10^8^ t and 8.92 × 10^8^ t, showing a continuous decrease. From 2000 to 2020, soil erosion decreases by 1.21 × 10^8^ t in the study area, which represents a decrease of 11.94%. In terms of the area occupied by each soil class (Table 3), with an increase in the erosion class, the area occupied by each class shows a decreasing trend. The data from 2000, 2010, and 2020 show the largest area occupied by very slight erosion, the proportion of which reaches 83.68%, 86.35%, and 86.69%, respectively. The area proportion (AP) occupied by slight erosion (LE) shows an increasing trend. The AP occupied by LE and moderate erosion shows a decreasing trend. During 2000–2020, the AP occupied by LE and intense erosion shows an increase over time, with an increase of 3.6% and 8.5%. The areas with slight, moderate, strong, and very strong erosion show a decrease over time, among which the greatest decrease is 29.72% for moderate erosion.

In Figure 3, the spatial variation in SE intensity in the study area between 2000 and 2020 is pronounced. In 2000, very slight erosion was predominantly distributed in Shandong, Henan, southern Shaanxi, northwestern Qinghai, and Erdos and Alxa in Inner Mongolia. Slight erosion and moderate erosion were majorly dispersed in the middle and upper reaches (URs), northern Shaanxi, southern Ningxia, and eastern Gansu. Strong erosion and severe erosion were predominantly dispersed in the URs of the YRB. In 2010, the spatial distribution of the erosion classes was relatively similar, but moderate erosion, strong erosion, and severe erosion in the middle and lower reaches of the study area were substantially reduced. Severe erosion and strong erosion in the Qinghai region in the upper reaches were enhanced. In 2020, SE intensity was further reduced, while moderate erosion and strong erosion in the central part of the study area had been substantially reduced. However, strong erosion in the middle reaches of northern Inner Mongolia was projected to have increased.

According to the calculation of SE class transfer in the study area (Table 4), the total alteration in SE between 2000 and 2020 is 323,145.23 km^2^. The transfer into very slight erosion and severe erosion is greater than the transfer out. The transfer into slight, moderate, strong, and very strong erosion is less than the transfer out, with the change in very strong erosion depicting the least change. Regarding transfer in, the largest area transferring from slight to very slight erosion is 117,494.06 km^2^. The area transferring from severe, very strong, and strong erosion to very slight erosion is 1286.93 km^2^, 3337.19 km^2^, and 6904.46 km^2^, respectively. Regarding transfer out, the largest area transferring from moderate to strong erosion is 6464.73 km^2^. The area transferring from very strong to severe erosion is the largest (2349.15 km^2^), followed by transferring to very slight erosion, which is 1069.75 km^2^. Therefore, soil and water conservation in areas of slight erosion cannot be neglected.

The transfer in (151,194.46 km^2^) of the very slight erosion class is greater than the transfer out (88,907.54 km^2^), and the turn-in area is 1.7 times larger than the transfer-out area. The transfer-in area makes up 46.78% of the total change area. This indicates that the soil erosion change is decreasing. However, the transfer in is also greater than the transfer out, indicating that the most severe soil erosion class is increasing.

### 3.2. Links between SE and Land-Usage Alteration

The transfer matrix model and the ArcGIS 10.2 software were used to obtain the land-usage alteration and transfer matrix for the YRB using data derived from 2000 to 2020 (Table 5).

In Table 4, the total land-use-change area between 2000 and 2020 reaches 631,789 km^2^, making up 31.12% of the total acreage. Cropland and unused land generally exhibited a dropping trend, and the other four land-use types presented an increasing trend. Between 2000 and 2020, the cropland area decreased from 367,435 km^2^ to 349,076 km^2^, representing a decrease of 18,359 km^2^, which is a decrease of 5%. The unused land area decreased by 29,742 km^2^, the woodland area increased by 4473 km^2^, the grassland area increased by 14,091 km^2^, the water area increased by 9300 km^2^, and the built-up land acreage rose by 20,747 km^2^. The built-up land acreage rose the most, whereas unused land decreased the least. Regarding the transfer out of each site, the farmland conversion was predominantly to grassland. The grassland conversion was the most extensive, with the land conversion mainly being to arable and unused land. The unused land conversion was mainly to grassland and watershed areas. The conversion of woodland was predominantly to grassland and cropland. The conversion of built-up land was mainly to grassland, and watershed was predominantly converted to grassland. In terms of the spatial extent of conversion, the largest spatial area for conversion was for grassland, which was predominantly converted from unused land. Cropland, forestland, watershed, and unused land were all mainly converted from grassland, and the built-up area was predominantly transformed from cropland. Therefore, “cropland–grassland,” “forestland–grassland,” and “unused–grassland” were the main types of land-usage change in the YRB between 2000 and 2020.

The spatial alterations in the investigation region between 2000 and 2020 were mainly dominated by cropland, woodland, grassland, and unused land. In general (Figure 4), the cropland area is predominantly dispersed in the middle and lower reaches of the YR, especially in Shandong and Henan. The woodland is predominantly distributed in the western Luoyang and Sanmenxia cities in Henan, the Lvliang Mountains and Taihang Mountains in Shanxi, southwest of Gansu, and the Yan’an in Shaanxi and Sichuan. The grassland and unused land are predominantly distributed in Qinghai and Inner Mongolia.

As indicated in Figure 5, different land uses in the study area present different erosion characteristics over the 20 years. In 2000, the average soil erosion for grassland and unused land in the study area were 7.06 t hm a^−2^ a^−1^ and 6.3 t hm^−2^ a^−1^, and LE had occurred. Soil erosion of cropland and woodland were both small 5 t hm^−2^ a^−1^, where very slight erosion had occurred. Grassland had the largest area for all five erosion classes, indicating that grassland constituted the land-usage type in the Yellow River basin prone to erosion in 2000. In 2010, soil erosion in cropland, woodland, and grassland decreased compared to 2000, and unused land cover increased with no change in the erosion class. During this period, grassland represented the largest area for all erosion classes except for the very strong class, which was slightly lower than the unused area. In 2020, soil erosion in cropland and woodland was further declined. Compared to 2010, soil erosion in grassland increased and soil erosion in unused land declined.

Soil erosion in cropland and woodland showed a continuous decline during 2000–2020 (Table 6), while soil erosion in grassland initially decreased and then increased, and soil erosion in unused land first increased and then decreased.

Briefly, by calculating the alterations in SE and different land-usage areas between 2000 and 2020 (Figure 6), a direct link exists between farmland and soil erosion. The decrease in cropland area reduces soil erosion and there is an inverse relationship between woodland and grassland. Here, an increase in spatial area reduces soil erosion, and a decrease in unused land causes an increase in soil erosion. Regarding specific change rates, cropland decreased by 5.0% while its soil erosion decreased by 55.46%, woodland increased by 2.31% while its soil erosion decreased by 42.34%, and grassland increased by 1.70% while its soil erosion decreased by 13.36% over the 20 years. Combined with the land-use matrix (Table 4), the transfer into woodland and grassland was greater than the transfer out. This indicates that the measures implemented by the Chinese government to return farmland to forest and grass have effectively reduced soil erosion.

### 3.3. Analyses of Future SE under Various Scenarios

In this study, in line with the development probability of various land usage in the investigation region in 2000, we predicted land-usage spatial distribution in 2020. The simulation results were tested using Kappa coefficients, and the Kappa coefficients of the natural change scenario and the ecological protection scenario were 0.873 and 0.881, respectively, indicating that the simulation is effective. The land-usage spatial distribution in the investigation region can be predicted for 2040, and the land-usage changes under the natural development and ecological protection scenarios can be obtained (Figure 7). We used the RUSLE model for calculating SE in the YRB under the two scenarios in 2040.

According to Figure 7 and Table 7, under the natural development scenario, the ratio of the acreage of all land types is grassland (41.74%) > unused land (25.06%) > cropland (17.07%) > woodland (10.05%) > construction land (3.42%) > water area (2.67%). Under the ecological protection scenario, the ratio of the area of land types is grassland (41.85%) > unused land (24.91%) > cropland (17.04%) > woodland (10.14%) > construction land (3.38%) > water area (2.67%). The implementation of strict conservation policies for cropland and grassland can increase woodland and grassland by 1978 km^2^ and 2407 km^2^, respectively. The SE spatial patterns in the two scenarios do not change substantially. Soil erosion is 4.81 t hm^−2^ a^−1^ in 2040 for the natural development scenario and 4.78 t hm^−2^ a^−1^ for the ecological protection scenario, both of which have a slight erosion grade, showing a 6.24% reduction in soil erosion under the ecological protection scenario compared to the natural development scenario. This indicates that soil erosion can be effectively reduced by changing land use through anthropogenic activities. However, the effect on the distribution of the spatial pattern for soil erosion is less pronounced.

## 4. Discussion

### 4.1. Major Finding and Result Comparison

Soil erosion poses a severe threat to surface processes and the ecological environment, and the YR, as the fifth largest river worldwide, has a wide catchment area, complex topography, fragile ecological environment, and severe soil erosion. This severely threatens the ecological security and high-quality development in the YRB [56]. In this study, we evaluated SE in the Yellow River Basin between 2000 and 2020 and found that the SE classification was all very slight erosion, and the actual erosion amounts were 10.13 × 10^8^ t, 9.20 × 10^8^ t and 8.92 × 10^8^ t (Table 2), showing continuous reduction. Soil erosion decreased by 11.92% from 2000 to 2020, which was closely related with the program of converting farmland to grassland conducted by the Chinese government in 1999, effectively reducing soil erosion by changing land-use types and increasing the proportion of woodland and grassland [57,58]. The implementation of these projects has changed the surface environment, and the increase of forest and grassland area will effectively reduce water and soil loss.

The main land types are grassland, unused land, cropland, and woodland in the YRB, and the transformation between the four is clearly defined, with cropland and unused land areas showing decreasing trend and woodland and grassland showing increasing trend. It aligns with the outcomes in Li et al. [59]. China invested a total of USD 378.5 billion in sustainable development projects targeting land systems in 1978–2015 [60] and forest ecological conservation. Grassland system projects, such as the restoration of forest ecology and grassland systems, reduced soil erosion by 12.9% in China from 2000 to 2010 [61]. In the Loess Plateau, large-scale restoration and afforestation of cropland and barren land have reduced soil erosion to historically low levels [62], and agro-industrial development has considerably improved the natural environment and quality of life in rural areas. The supplementary consolidation of valley-bottom cropland in China has alleviated the pressure of agricultural development on sloping land and reduced soil erosion by a further 10% [63]. It has promoted the sustainable development of human and natural systems, reflecting the achievements of China’s environmental policies and environmental governance [64,65]. Given that it is the most pronounced factor influenced by human activities, land-usage type has a considerable impact on SE through its variability [66]. Patch areas encompassing various land-usage types feature different effects on SE, with a positive correlation between patch areas for cropland and sediment production and a negative correlation between patch areas for forest and grassland and sediment production [67]. We found a positive relationship between cropland and soil erosion, with a decrease in cropland reducing soil erosion. In 20 years, cropland has decreased by 5.0% while its soil erosion has decreased by 55.46%; forestland has increased by 2.31% while its soil erosion has decreased by 42.34%; and grassland has increased by 1.70% while its soil erosion has decreased by 13.36% in the YRB (Figure 6). There is an inverse relationship between forest and grassland, where a rise in forest area reduces SE and a drop in unused land causes an increase in soil erosion.

### 4.2. Land-Use Change and Soil Erosion under Different Scenarios in the Future

Land-use simulation and prediction have been a key focus for GIS researchers and policy makers. We can apply models to simulate future land-use layouts to replace the complex correlations and feedback relationships of different land uses [68]. Liang et al. (2021) proposed a model based on a future land-usage simulation model with the PLUS model in line with the growth probability output from the land expansion analytical strategy [53]. This can better simulate the patch growth of multiple land-usage types at fine scales and can generate more realistic landscape patterns. We predicted land-usage changes in line with the PLUS model for two scenarios, namely those of natural development and ecological protection in the YRB in 2040. SE in 2040 was calculated in line with the projected land usage, and the SE in 2040 was found to be 4.81 t hm^−2^ a^−1^ for the natural scenario and 4.78 t hm^−2^ a^−1^ for the ecological protection scenario, both being a slight-erosion class. In comparison to the natural development scenario, SE under the ecological protection scenario decreases by 6.24%, further indicating that soil erosion can be effectively reduced by implementing forest and grassland conservation policies. Wei et al. analyzed the links between SE and fallowing in the Fangta watershed and found that the fallowing project implemented in the watershed was the major cause of the drop in SE [69].

### 4.3. Limitations of This Study

The main limitation of this study is that it only predicted the *P*-factor, rather than the *R*-factor and *C*-factor. As the main factor of SE, the *R*-factor has a greater impact. Therefore, future assessments of soil erosion are not sufficiently accurate. We used data with a spatial resolution of 1 km, which is relatively low, and soil erosion modeling is affected by the size of the resolution. The higher the resolution, the more accurate the SE assessment [70]. Therefore, higher resolution data should be considered for soil erosion calculations in the future. Despite some limitations, this study provides reliable information about past and future SE in the YRB, for example, the areas featuring severe SE are majorly distributed in the middle and upper reaches, and an increase in woodlands and grasslands can effectively reduce soil erosion.

## 5. Conclusions

Our research mainly discussed the relationship between land-use change and soil erosion in the YRB from 2000 to 2020, revealed the impact of land-use change on soil erosion, and studied the characteristics of soil erosion under different scenarios in the YRB in the future.

We found that soil erosion in the YRB from 2000 to 2020 was slight, and the main land-use types were grassland, unused land, cropland, and woodland. Soil erosion is significantly mitigated. There is a direct relationship between cropland and soil erosion, with a decrease in the cropland area reducing soil erosion. There is an inverse relationship between woodland and grassland, where an increase in woodland area reduces soil erosion. Woodland has the best effect in terms of mitigating soil erosion. In 2040, compared to the natural development scenario, the ecological protection scenario in the Yellow River basin can increase forestland and grassland by 1978 km^2^ and 2407 km^2,^ respectively, and soil erosion will decrease by 6.24%. Without considering climate change, soil erosion pressure will increase in the future, but the ecological protection scenario can effectively reduce soil erosion. The study area has a large proportion of unused land, so it is recommended to increase the development of unused land and transform it into forested grassland. According to the topographic conditions, some grasslands should be converted into forested land, increase the proportion of forested land, and measures should be strengthened, for example, by sloping cropland improvement. Our conclusions provide a scientific basis for future land-use planning, as well as soil and water loss management and prevention, in the Yellow River Basin.

## Figures and Tables

**Figure 1 ijerph-20-01222-f001:**
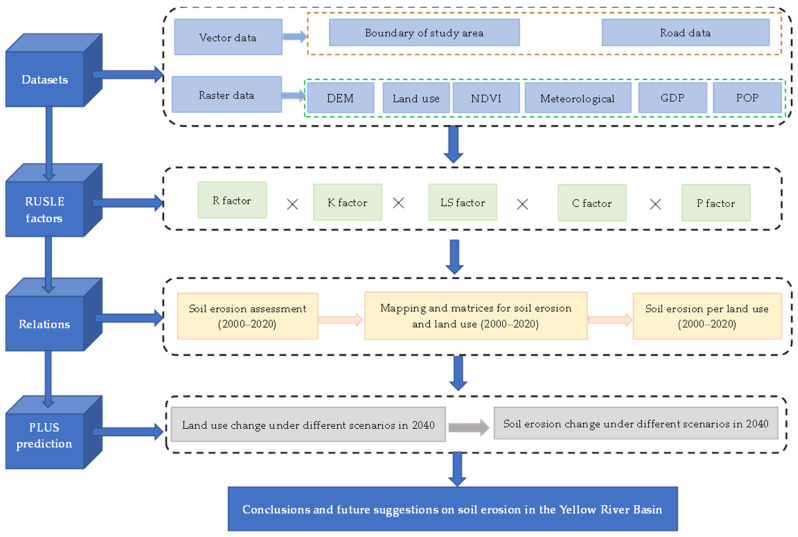
Research framework.

**Figure 2 ijerph-20-01222-f002:**
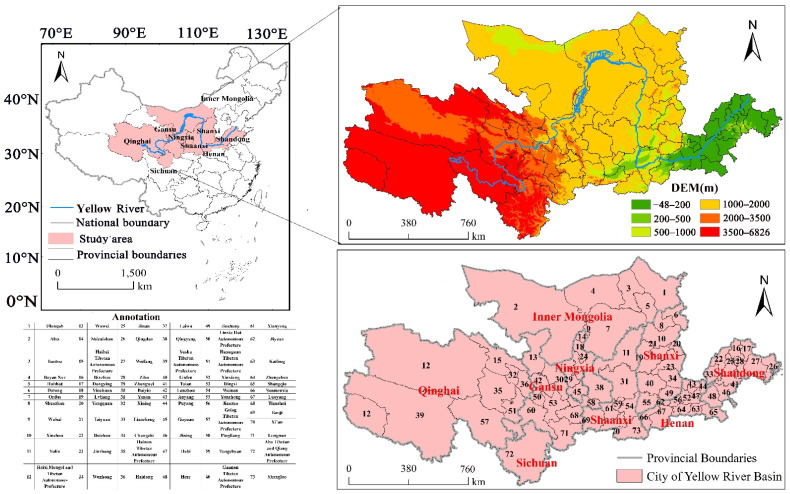
Elevation map of the investigation region.

**Figure 3 ijerph-20-01222-f003:**
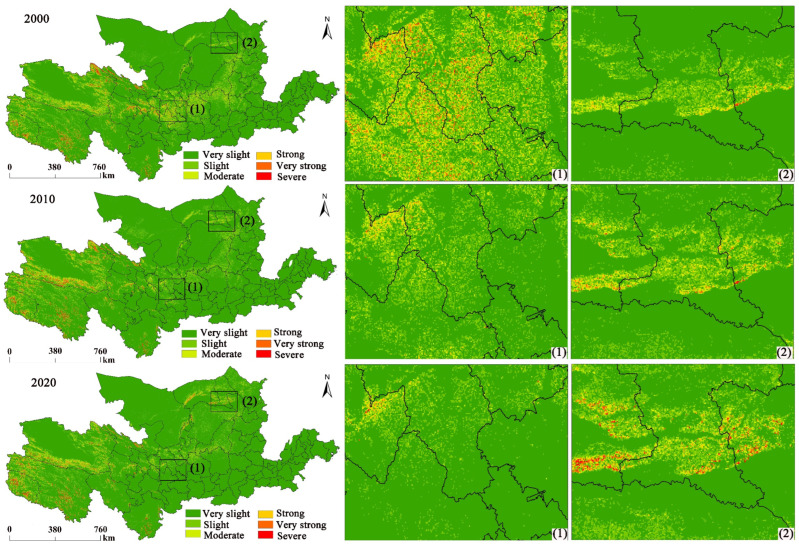
SE spatial distribution from 2000 to 2020. (1) and (2) represent two randomly selected areas of significant change.

**Figure 4 ijerph-20-01222-f004:**
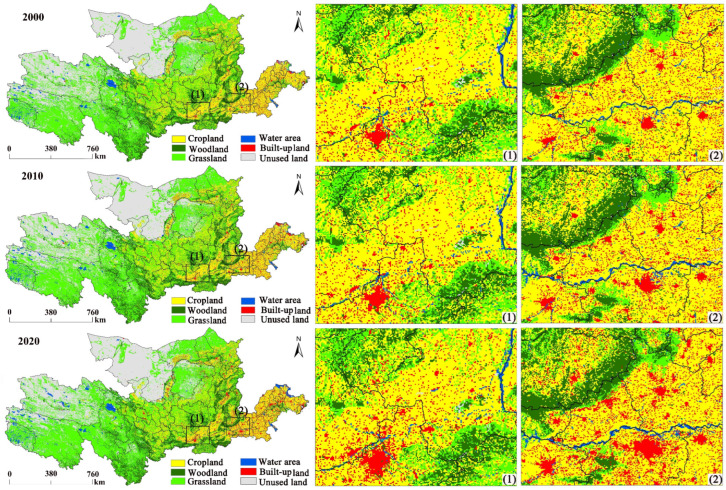
Spatial change of land use in the YRB between 2000 and 2020. (1) and (2) represent two randomly selected areas of significant change.

**Figure 5 ijerph-20-01222-f005:**
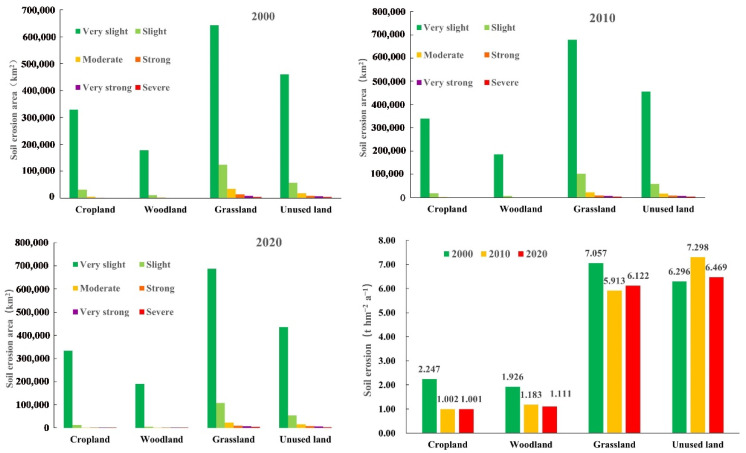
Soil erosion acreage and numerical value of various land-usage types between 2000 and 2020.

**Figure 6 ijerph-20-01222-f006:**
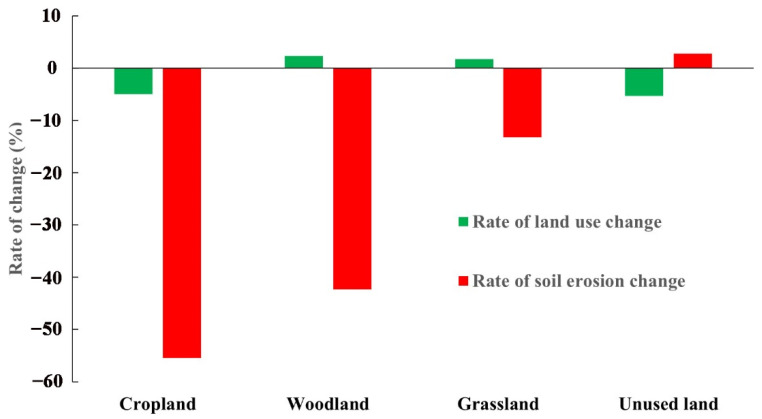
Links between SE and land-usage alteration from 2000 to 2020.

**Figure 7 ijerph-20-01222-f007:**
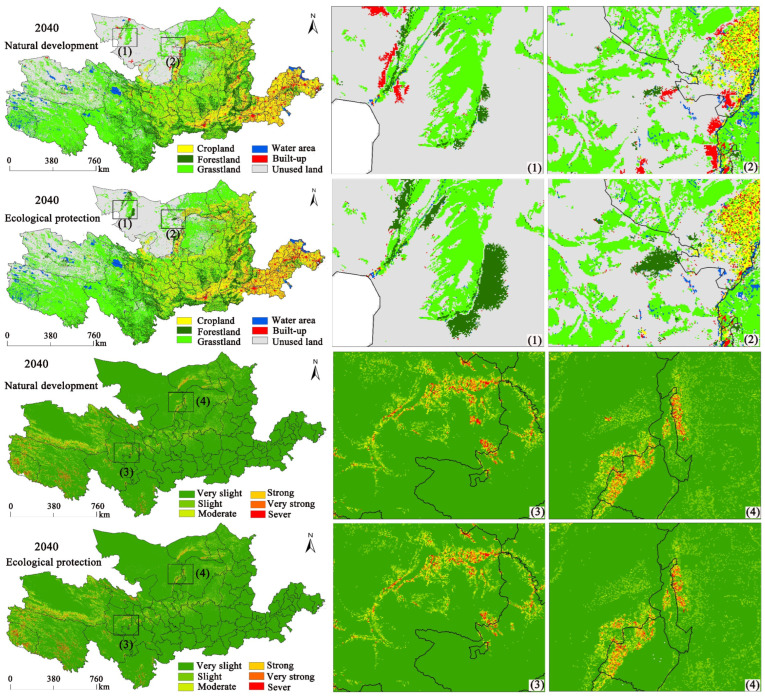
Maps of SE and land usage under natural development and ecological protection in 2040. (1), (2), (3) and (4) represent randomly selected areas of significant change.

**Table 1 ijerph-20-01222-t001:** List of datasets and their original sources.

Data Type	Data Source	Resolution Ratio	Time
DEM	https://earthexplorer.usgs.gov/, accessed on 1 April 2022	30 m	
LUCC	https://www.resdc.cn, accessed on 1 April 2022	1 km	2000–2020
Soil	https://www.fao.org/land-water/databases-and-software/hwsd/en/, accessed on 5 April 2022	1 km	
Meteorological	https://data.cma.cn/, accessed on 1 April 2022	1 km	2000–2020
NDVI	https://modis.gsfc.nasa.gov/, accessed on 5 April 2022	250 m	2000–2020
Road	https://www.openstreetmap.org/, accessed on 5 April 2022	1 km	2020
POP	https://www.resdc.cn/, accessed on1 April 2022	1 km	2019
GDP	https://www.resdc.cn/, accessed on1 April 2022	1 km	2019

**Table 2 ijerph-20-01222-t002:** Characteristics of different soil erosion intensities from 2000 to 2020.

Year	Soil Erosion (t hm^−2^ a^−1^)	Total Soil Erosion (10^8^ t)
2000	5.28	10.13
2010	4.80	9.20
2020	4.65	8.92

**Table 3 ijerph-20-01222-t003:** Characteristics of different soil erosion intensities from 2000 to 2020.

Classification of SE	2000			2010			2020		
	Area (10^6^hm^2^)	SE (t hm^−2^ a^−1^)	Amount of SE (10^8^ t)	Area (10^6^hm^2^)	SE (t hm^−2^ a^−1^)	Amount of SE (10^8^ t)	Area (10^6^hm^2^)	SE (t hm^−2^ a^−1^)	Amount of SE (10^8^ t)
Very slight	172.49	1.00	1.63	178.01	0.97	1.64	178.72	0.57	0.95
Slight	22.76	11.53	2.11	19.25	11.20	1.64	18.82	11.14	1.95
Moderate	6.09	34.81	1.85	4.44	34.90	1.34	4.28	35.19	1.40
Strong	2.42	62.24	1.35	1.92	62.67	1.08	1.93	62.92	1.13
Very strong	1.58	106.09	1.52	1.61	107.82	1.57	1.53	107.88	1.53
Severe	0.80	223.81	1.66	0.93	225.78	1.93	0.87	243.22	1.96

**Table 4 ijerph-20-01222-t004:** Soil erosion transfer matrix for the Yellow River Basin derived from data from 2000 to 2020 (km^2^).

Classification of SE	Very Slight	Slight	Moderate	Strong	Very Strong	Severe	Transfer Out
Very slight	-	78,864.79	5367.03	1988.98	1616.99	1069.75	88,907.54
Slight	117,494.06	-	19,721.04	4954.18	2016.94	654.75	144,840.97
Moderate	22,172.32	17,240.72	-	6464.73	3717.79	722.48	50,318.04
Strong	6904.46	5685.27	3966.14	-	3571.57	1012.77	21,140.20
Very strong	3337.19	2844.78	2306.15	1973.93	-	2349.15	12,811.20
Severe	1286.93	814.95	757.96	841.82	1425.62	-	5127.28
Transfer in	151,194.96	105,450.50	32,118.31	16,223.65	12,348.90	5808.91	323,145.23

**Table 5 ijerph-20-01222-t005:** Land-usage matrix for the Yellow River Basin 2000–2020 (km^2^).

LUCC	Cropland	Woodland	Grassland	Water area	Built-Up Land	Unused Land	Transfer Out	Total Area in 2020
Cropland	-	18,040	75,381	5879	35,688	3472	138,460	349,076
Woodland	16,329	-	52,217	973	1974	3090	74,583	197,962
Grassland	72,178	56,200	-	12,873	8216	87,530	236,997	841,934
Water area	4815	803	10,387	-	1242	5525	22,772	53,999
Built-up land	21,537	788	3639	1945	-	509	28,418	61,650
Unused land	5256	3482	109,565	10,271	1985	-	130,559	525,281
Transfer in	120,115	79,313	251,189	31,941	49,105	100,126	-	
Total area in 2000	367,435	193,489	827,843	44,699	40,903	555,023		-

Note: Cropland refers to the land where crops are planted, including mature cultivated land, newly developed wasteland, leisure land, alternate land, and grass–crop rotation land; land for agricultural fruit, mulberry, and forestry mainly for planting crops; and beach and seashore cultivated for more than three years. Woodland refers to forestland where trees, shrubs, bamboos, and coastal mangroves grow. Grassland refers to all kinds of grassland mainly growing herbaceous plants with coverage of more than 5%, including shrub grassland mainly for grazing and sparse forest grassland with canopy density of less than 10%. Built-up land refers to urban and rural residential areas and industrial, mining, transportation, and other land outside. Unused land refers to land that has not been used at the present, including land that is difficult to use, such as sandy land, gobi, saline alkali land, marsh land, bare land, bare rocky land, alpine desert, and tundra.

**Table 6 ijerph-20-01222-t006:** Proportion of different land-use areas under different soil erosion grades in the YRB from 2000 to 2020.

Classification of SE	LUCC	Proportion of Soil Erosion Area (%)	
		2000	2020
Very slight	Cropland	89.53	95.38
	Woodland	92.06	96.10
	Grassland	77.62	81.66
	Unused land	82.92	82.97
Slight	Cropland	8.48	3.75
	Woodland	5.85	2.76
	Grassland	14.98	12.83
	Unused land	10.38	10.40
Moderate	Cropland	1.35	0.54
	Woodland	1.33	0.58
	Grassland	4.22	2.71
	Unused land	3.21	3.00
Strong	Cropland	0.41	0.15
	Woodland	0.36	0.22
	Grassland	1.62	1.16
	Unused land	1.51	1.55
Very strong	Cropland	0.10	0.05
	Woodland	0.16	0.17
	Grassland	1.00	0.93
	Unused land	1.20	1.25
Severe	Cropland	0.01	0.01
	Woodland	0.06	0.06
	Grassland	0.48	0.60
	Unused land	0.69	0.62

**Table 7 ijerph-20-01222-t007:** Proportion of land-use area and soil erosion characteristics under different scenarios in the YRB in 2040.

LUCC	Natural Development Scenario	Ecological Protection Scenario
Cropland	17.07%	17.04%
Woodland	10.05%	10.14%
Grassland	41.74%	41.85%
Water area	2.67%	2.67%
Built-up land	3.42%	3.38%
Unused land	25.06%	24.91%
Soil erosion	4.81 t hm^−2^ a^−1^	4.78 t hm^−2^ a^−1^

## Data Availability

Not applicable.

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
