# Peer review of "Soil Erosion Characteristics and Scenario Analysis in the Yellow River Basin Based on PLUS and RUSLE Models"

_ijerph, 2023, doi:10.3390/ijerph20021222_

Round 1

Reviewer 1 Report

Please clearly state and define the objective of this study, and the methods used in predicting SE in YRB, as well as the limitation of the methods used in this study. Was ground measurement used in predicting SE? If this study only based on modeling using GIS with low resolution spatial data and only scenarios, then the author should use more “if” and “will be” or “would be” not past tense sush as …increased by 1,978 km2, etc. because it was not based on ground measurement.

The limitation of the methods used and the results have to be clearly stated in the abstract (which are not appear) so that the readers & policy makers will understand what this paper is trying to present. In addition, GIS modeling has to incorporate ground measurement for model calibration and to bring modeling results to as close as possible to the real conditions on the ground.

Author Response

Point–to–point responses

Reviewer #1

Commented 1: Please clearly state and define the objective of this study, and the methods used in predicting SE in YRB, as well as the limitation of the methods used in this study. Was ground measurement used in predicting SE? If this study only based on modeling using GIS with low resolution spatial data and only scenarios, then the author should use more “if” and “will be” or “would be” not past tense sush as …increased by 1,978 km2, etc. because it was not based on ground measurement.

Response: Thanks for your comments. We have revised the full text based on your suggestions. We have added objectives of this study at the end of the introduction. We have added PLUS model and simulation process in the method part, too. This model is used to predict and evaluate the soil erosion of the Yellow River basin in 2040 by correlating with the P factor in the RUSLE model, predicting land use change and combining P factor. The limitations of the method have been mentioned in the abstract and discussed in detail in the discussion。Our study based on modeling using GIS with low resolution spatial data and only scenarios, we use “will be” replace past tense in the full text. Please see the content in lines 120-128, 246-294, 523-532, as follows:

  1. Introduction

Our specific objectives are: (1) to reveal the changes of soil erosion in the YRB from 2000 to 2020, and the characteristics of changes under different scenarios; (2) to rela-tionship between land use change and soil erosion was discussed, and the impact of land use change on soil erosion was analyzed quantitatively; (3) based on PLUS model, the land use change is predicted, and the soil erosion in the YRB in 2040 is predicted in combination with the P-factor of RUSLE model. It reveals the changes in SE under different scenarios, supplies a scientific basis for more science driven erosion control, ecological zoning, and coordinated improvement of the economic environment in the YRB.

2.5 PLUS model

Previous prediction models have played a limited role in exploring the causes of land use change, and it is difficult to dynamically simulate patch level changes of multiple land use types in time and space, especially for forest land, grassland and other natural land use types. The PLUS mode is a new model constructed in line with meta-cellular automata [56]. By analyzing the spatial characteristics of land usage expansion parts and drivers between the two phases of land use data, the random forest algorithm is employed for sampling and calculating land usage expansion for obtaining development probabilities of various types of land usage one by one; then the comprehensive probability of land usage alteration is obtained in line with adap-tive inertial competition mechanisms of roulette; finally, the random patch generation, transition matrix and threshold are combined. The decrement mechanism realizes op-timization and determines the final land use method. Compared with the simulations of various models commonly used in current related research, the PLUS mode fea-tures a higher simulation accuracy [57]. The simulation results can better support re-lated studies on future land-use changes. The main advantages of this model are: (1) A new analysis strategy can be used to better explore the incentives of various land use changes. (2) It includes a new multi category seed growth mechanism, which can bet-ter simulate the patch level changes of multi category land use. (3) Coupled with the multi-objective optimization algorithm, the simulation results can better support the planning policy to achieve sustainable development [58].

The PLUS model prediction scenario settings and processes are as follows:

(1) We extract the land use expansion from 2000 to 2020 according to the land expansion module in PLUS software. Then, combined with the land expansion analy-sis strategy module, 12 driving factors are selected, including elevation, slope, aspect, annual average precipitation, annual average temperature, GDP, population, distance from expressway, distance from railway, distance from main roads, distance from water area and soil type, to generate the development probability of each type of land.

(2) Based on the CA model module based on multi class random patch seeds in PLUS software, the land development probability obtained in the previous step is taken as the basic condition, and then the water area is set as the restricted develop-ment condition to obtain the diffusion coefficient, and then the domain weight is cal-culated according to the proportion of the expansion area of each land type. At the same time, two change scenarios of natural development and ecological protection have been established.

Scenario of natural development. It implies the trend and charac-teristics of land use change with no significant change in the factors affecting land use change during 2000–2020. The land use structure by 2040 was predicted according to the transfer probability from 2000 to 2020.

Scenario of ecological protection. According to the requirements of ecological protection, the transfer of forest land, grassland, and water areas would be strictly controlled. Although grassland can be converted into forest land and forest land can be converted into grassland, it cannot be converted into other land types.

(3) Select the year 2000 as the base start time. Use the development probability of each land use from 2000 to 2020 to predict the land use in 2020. The verification mod-ule in PLUS software is used to input the actual land use data and predicted land use data in 2020, and its results are verified by Kappa coefficient to evaluate the simula-tion results. On this basis, further simulate the land use under the natural develop-ment scenario and ecological protection scenario in 2040. The simulation results are used to calculate the P factor of RUSLE, because this factor is greatly affected by hu-man activities. By predicting land use, and then predicting soil erosion under the in-fluence of human activities, the relationship between land use change and soil erosion is discussed.

4.3 Limitations of this study

The main limitation of this study is that only predicted the P-factor, rather than the R-factor and C-factor. As the main factor of SE, the R-factor has a greater impact. Therefore, future assessments of soil erosion are not sufficiently accurate. We used data with a spatial resolution of 1 km, which is relatively low, and soil erosion mod-eling will be affected by the size of the resolution. The higher the resolution, the more accurate the SE assessment [73]. Therefore, higher resolution data should be consid-ered for soil erosion calculations in the future. Despite some limitations, this study provides reliable information about past and future SE in the YRB, for example, the areas featuring severe SE are majorly distributed in the middle and upper reaches, and the increase in woodlands and grasslands can effectively reduce soil erosion.

Commented 2: There has to be a valid reference for Line137 to Line 139

Response: Thank you for your suggestions, we have added a valid reference. Please see the reference in lines 634-635, reference as follows:

  1. Jiang, W.G.; Yuan, L.H.; Wang, W.J.; Cao, R.; Zhang, Y.F.; Shen, W.M. Spatio-temporal analysis of vegetation variation in the Yellow River Basin. Ecol Indic. 2015, 51, 117-126.

Commented 3: Which software of the GIS used? Please meantion.

Response: Thank you for your comments. We have modified the specific version of geographic information system (GIS). Please see the content in lines 162-164, as follows:

Datasets were obtained online with a resolution of 1 × 1 km, and for maintaining data resolution consistency, data higher than 1 km resolution were uniformly resampled using ArcGIS 10.2 software.

Commented 4: What are C & P?

Response: Thank you for your comments. We have added the meaning of C & P. Please see the content in lines 177-178, as follows:

C indicate the land cover and management factor (dimensionless); and P indicate the conservation practice factor (dimensionless).

Commented 5: Please provide references to this statement.

Response: Thank you for your comments. we have added a reference. Please see the content in lines 657-658, reference as follows:

  1. Cunha,E.R.D.; Santos,C.A.G.; Silva, R.M1.D.; Panachuki, E.; Oliveira,P.T.S.D.; Oliveira, N.D.S.; Falcão, K.D.S. Assessment of current and future land use/cover changes in soil erosion in the Rio da Prata basin (Brazil). Sci Total Environ. 2022, 818, 151811.

Commented 6: Please provide references to the statement in Line 175-177.

Response: Thank you for your comments. We have added a reference. Please see the content in lines 659-660, reference as follows:

  1. Panagos, P.; Meusburger, K.; Ballabio, C.; Borrelli, P.; Alewell, C. Soil erodibility in Europe: a high-resolution dataset based on LUCAS. Sci. Total Environ. 2014, 479, 189–200.

Commented 7: It does not always accelerate SE.

Response: Thank you for your suggestions. We agree your opinion. Regional topography does not always accelerate SE. We have revised the acceleration to affect. Please see the content in line 208, as follows:

Regional topography can affect erosion caused by rainfall.

Commented 8: Isn’t slope length also a slope factor?

Response: Thank you for your suggestions. Slope length is a slop factor. We have revised the slope to steepness. Please see the content in line 210-211, as follows:

Slope length and steepness factors are important topographic parameters for estimating soil erosion in the RUSLE type.

Commented 9: By?

Response: Yes, it should be by, not through. We have revised the through to by. Please see the content in line 211-212, as follows:

L-values are computed utilizing the classical equation presented by Wischmeier et al.

Commented 10: Should not use this word, because in relation to topography, aspect means direction of the slope (the values are between 0 and 360 degrees)

Response: Thank you for your suggestions. We have revised the aspects to affects. Please see the content in lines 225-226, as follows:

and the magnitude of the C value is influenced by affects such as vegetation growth, vegetation type, and cover.

Commented 11: It should be specified what variables of the growth.

Response: Thank you for your suggestions. We have revised the growth to vegetation growth. Please see the content in lines 225-226, as follows:

and the magnitude of the C value is influenced by affects such as vegetation growth, vegetation type, and cover.

Commented 12: Is it from 2000 to 2020?

Response: Thank you for your suggestions. We have revised this sentence. Please see the content in line 241 as follows:

We used LUCM to study the transfer in and out of each land type from 2000 to 2020 [55].

Commented 13: This values cannot be found in Table 2?

Response: Thank you for your comments. We have added the Table 2. Please see the content in lines 298-299, as follows:

By calculating the soil erosion of the YRB from 2000 to 2020 and calculating the specific amount of soil erosion, the results of SE changes in the study area are shown in Table 2.

Table 2. Characteristics of different soil erosion intensities from 2000–2020

Year

Soil erosiont hm−2 a−1

Total soil erosion (108 t)

2000

5.28

10.13

2010

4.80

9.20

2020

4.65

8.92

Commented 14: Where is the three points of data from, are they form ground measurements so that the authors say “ actual erosion”

Response: Thank you for your comments. Three points of data from Table 2. We have added the Table 2. “actual erosion” revised as “total soil erosion”. Please see the content in lines 309-310, as follows:

The total soil erosion is 10.13×108 t, 9.20×108 t and 8.92×108 t, showing a continuous decrease.

Table 2. Characteristics of different soil erosion intensities from 2000–2020

Year

Soil erosiont hm−2 a−1

Total soil erosion (108 t)

2000

5.28

10.13

2010

4.80

9.20

2020

4.65

8.92

Commented 15: Where this value from? What was the erosion in 2000 and in 2020? Was it 10.12 - 8.92 = 1.20 equals to11.86%, not 11.92%? If Yes, these should appear in Table 2 because the authors say “According to Table 2”. Please also do the same for all tables so that the authors will not confuse the readers.

Response: Thank you for your comments. This value from Table 2. We have modified these values. 10.13 - 8.92 = 1.21 equals to 11.94%. Please see the content in lines 310-311, as follows:

From 2000 to 2020, soil erosion decreased by 1.21×108 t in the study area, which rep-resents a decrease of 11.94%.

Commented 16: Where is the three % in line 255 from? My calculation results are 11.04%; 9.34%; 9.13%; are they correct? If Yes, please also revise the calculations of data for the entire Results and all tables. Please make tables more complete and easier to follow by the readers in relation to the discussion, conclusion, and the implication suggested by the authors.

Response: Thank you for your suggestions. This value from Table 3. Our statement is wrong. It should be very slight erosion rather than erosion. The specific algorithm is

172.49/(172.40+22.76+6.09+2.42+1.58+0.8)=83.68%. We have modified these values. Please see the content in lines 313-315, as follows:

The data from 2000, 2010, and 2020 showed the largest area occupied by very slight erosion, the proportion of which reached 83.68%, 86.35%, and 86.69%, respectively.

Table 3. Characteristics of different soil erosion intensities from 2000–2020

Classification of

SE

2000

2010

2020

Area

(106hm2)

SE

(t·hm−2 a−1)

Amount of SE (108·t)

Area (106hm2)

SE

(t·hm−2 a−1)

Amount of SE(108·t)

Area (106hm2)

SE (t·hm−2 a−1)

Amount of SE(108·t)

Very slight

172.49

1.00

1.63

178.01

0.97

1.64

178.72

0.57

0.95

Slight

22.76

11.53

2.11

19.25

11.20

1.64

18.82

11.14

1.95

Moderate

6.09

34.81

1.85

4.44

34.90

1.34

4.28

35.19

1.40

Strong

2.42

62.24

1.35

1.92

62.67

1.08

1.93

62.92

1.13

Very strong

1.58

106.09

1.52

1.61

107.82

1.57

1.53

107.88

1.53

Severe

0.80

223.81

1.66

0.93

225.78

1.93

0.87

243.22

1.96

Commented 17: Please add legends in line 275 what was (1) and (2)

Response: Thank you for your comments. Due to the large area of the Yellow River basin, the change of soil erosion is not very obvious from the whole map. Therefore, we randomly selected two regions with obvious changes as the enlarged map, indicating that soil erosion has changed from 2000 to 2020. We have added legends about (1) and (2). Please see the content in line 334, as follows:

Figure 3. SE spatial distribution from 2000 to 2020. (1) and (2) represent two areas of significant change randomly selected.

Commented 18: These terms are not in Table 3 but the values seem to come from Table 3?

Response: Thank you for your suggestions. These terms are in Table 4. We checked the terminology of the entire text and corrected it. Please see the content in lines 341-343, as follows:

Regarding transfer-in, the largest area of slight to very slight erosion is 117,494.06 km2. The area of severe, very strong, and strong to very slight is 1,286.93 km2, 3,337.19 km2, and 6,904.46 km2, respectively.

Table 4. Soil erosion transfer matrix for the Yellow River Basin derived from data from 2000–2020 (km2)

Classification of

SE

Very slight

Slight

Moderate

Strong

Very strong

Severe

Transfer out

Very slight

78,864.79

5,367.03

1,988.98

1,616.99

1,069.75

88,907.54

Slight

117,494.06

19,721.04

4,954.18

2,016.94

654.75

144,840.97

Moderate

22,172.32

17,240.72

6,464.73

3,717.79

722.48

50,318.04

Strong

6,904.46

5,685.27

3,966.14

3,571.57

1,012.77

21,140.20

Very strong

3,337.19

2,844.78

2,306.15

1,973.93

2,349.15

12,811.20

Severe

1,286.93

814.95

757.96

841.82

1,425.62

5,127.28

Transfer in

151,194.96

105,450.50

32,118.31

16,223.65

12,348.90

5,808.91

323,145.23

Commented 19: What does “The largest area of intense erosion is very intense” mean? Where is the evidence?

Response: Thank you for your comments. “intense erosion” should be “strong erosion ”“The largest area of intense erosion is very intense” should be “Very strong transfer in severe area is the largest area” These data are in Table 4. We checked the terminology of the entire text and corrected it. Please see the content in lines 344-345, as follows:

Very strong transfer in severe area is the largest area (2349.15 km2), followed by very slight erosion of is 1069.75 km2.

Table 4. Soil erosion transfer matrix for the Yellow River Basin derived from data from 2000–2020 (km2)

Classification of

SE

Very slight

Slight

Moderate

Strong

Very strong

Severe

Transfer out

Very slight

78,864.79

5,367.03

1,988.98

1,616.99

1,069.75

88,907.54

Slight

117,494.06

19,721.04

4,954.18

2,016.94

654.75

144,840.97

Moderate

22,172.32

17,240.72

6,464.73

3,717.79

722.48

50,318.04

Strong

6,904.46

5,685.27

3,966.14

3,571.57

1,012.77

21,140.20

Very strong

3,337.19

2,844.78

2,306.15

1,973.93

2,349.15

12,811.20

Severe

1,286.93

814.95

757.96

841.82

1,425.62

5,127.28

Transfer in

151,194.96

105,450.50

32,118.31

16,223.65

12,348.90

5,808.91

323,145.23

Commented 20: Please the same terms in the Table 3 and the explanation of the data in the text, and explain and define what the terms mean in relation to the data used. For example the classes in the table are strong, very strong, etc, while the authors discuss about “intense”and “very intense”. In 278-279, the authors use the terms “transter-in” & “transter-out”, and in Table 3, total transfer-in is in the last row while transfer-out is in the last column, but for example, for Slight erosion, the authors say that transfer-in (105,450.50) is greater that transfer-out (144,840.97); are the author confused or they confusing the readers. Then, in line 287 the authors talk about turn-in and turn-out???

Response: Thank you for your opinions. “intense” and “very intense” should be “strong ” and “very strong”. “turn-in” and “turn-out” should be “transter-in” & “transter-out”. For slight erosion, the statement is wrong, and it should be very slight erosion. We have checked the entire text and corrected it according to the terms in the table. Please see the content in lines 348-350, as follows:

The transfer in (151,194.46 km2) of the very slight erosion class is greater than the transfer out (88, 907.54 km2), and the turn-in area is 1.7 times larger than the transfer out area. The transfer in area makes up 46.78% of the total change area.

Commented 21: What is t? And what are the unit of the values in Table 3? Why the authors talk about km2 in line 277 to line 283 in discussing the values in Table 3?

Response: Thank you for your comments. This unit “t” statement is wrong, it should be "km2". So, we talk about km2 in table 4.

Table 4. Soil erosion transfer matrix for the Yellow River Basin derived from data from 2000–2020 (km2)

Classification of

SE

Very slight

Slight

Moderate

Strong

Very strong

Severe

Transfer out

Very slight

78,864.79

5,367.03

1,988.98

1,616.99

1,069.75

88,907.54

Slight

117,494.06

19,721.04

4,954.18

2,016.94

654.75

144,840.97

Moderate

22,172.32

17,240.72

6,464.73

3,717.79

722.48

50,318.04

Strong

6,904.46

5,685.27

3,966.14

3,571.57

1,012.77

21,140.20

Very strong

3,337.19

2,844.78

2,306.15

1,973.93

2,349.15

12,811.20

Severe

1,286.93

814.95

757.96

841.82

1,425.62

5,127.28

Transfer in

151,194.96

105,450.50

32,118.31

16,223.65

12,348.90

5,808.91

323,145.23

Commented 22: This table use different format of those values? Is comma in this table used for decimal point or as thousand separators? Data have to presented in a very accurate manner because they have to mean a specific thing very exactly.

Response: Thank you for your opinions. Comma in this table used for as thousand separators. We have modified the data in Table 5. Please see the content in lines 354-357, as follows:

The transfer matrix model and ArcGIS 10.2 software were used to obtain the land-usage alteration and transfer matrix for the YRB using data derived from 2000 to 2020 (Table 5).

Table 5. Land usage matrix for the Yellow River Basin 2000–2020 (km2)

LUCC

Cropland

Woodland

Grassland

Water area

Built-up land

Unused land

Transfer out

Total area in 2020

Cropland

18,040

75,381

5,879

35,688

3,472

138,460

349,076

Woodland

16,329

52,217

973

1,974

3,090

74,583

197,962

Grassland

72,178

56,200

12,873

8,216

87,530

236,997

841,934

Water area

4,815

803

10,387

1,242

5,525

22,772

53,999

Built-up land

21,537

788

3,639

1,945

509

28,418

61,650

Unused land

5,256

3,482

109,565

10,271

1,985

130,559

525,281

Transfer in

120,115

79,313

251,189

31,941

49,105

100,126

Total area in 2000

367,435

193,489

827,843

44,699

40,903

555,023

Commented 23: What is exactly the definition of unused land in this table, as well as arable land in the text.

Response: Thank you for your comments. Cropland refers to the land where crops are planted, including mature cultivated land, newly developed wasteland, leisure land, alternate land and grassland crop rotation land; Land for agricultural fruit, mulberry, and forestry mainly for planting crops; Beach and seashore cultivated for more than three years.

Built-up land refers to urban and rural residential areas and industrial, mining, transportation and other land outside.

Unused land refers to the land that has not been used at present, including the land that is difficult to use. Including sandy land, gobi, saline alkali land, marsh land, bare land, bare rocky land, alpine desert, tundra, etc.

The expression of arable land is wrong, and it has been changed to cropland in full text.

Commented 24: There is no “Arable land” in table4, so please define exactly what the authors mean with the term “arable land” for example by putting an explanation in the bracket, so that the perception of the authors can be the same as that of the readers. This is very crucial to be done throughout this paper so that the authors will not look like “confused” the readers.

Response: Thank you for your comments. “Arable land” should be “Cropland”. We have modified the cultivated land in the full text according to the contents in Table 5, and made remarks on the meaning of land use. Please see the content in lines 357-369, as follows:

Table 5. Land usage matrix for the Yellow River Basin 2000–2020 (km2)

LUCC

Cropland

Woodland

Grassland

Water area

Built-up land

Unused land

Transfer out

Total area in 2020

Cropland

18,040

75,381

5,879

35,688

3,472

138,460

349,076

Woodland

16,329

52,217

973

1,974

3,090

74,583

197,962

Grassland

72,178

56,200

12,873

8,216

87,530

236,997

841,934

Water area

4,815

803

10,387

1,242

5,525

22,772

53,999

Built-up land

21,537

788

3,639

1,945

509

28,418

61,650

Unused land

5,256

3,482

109,565

10,271

1,985

130,559

525,281

Transfer in

120,115

79,313

251,189

31,941

49,105

100,126

Total area in 2000

367,435

193,489

827,843

44,699

40,903

555,023

Note: Cropland refers to the land where crops are planted, including mature cropland, newly developed wasteland, leisure land, alternate land and grassland crop rotation land; Land for agricultural fruit, mulberry, and forestry mainly for planting crops; Beach and seashore cultivated for more than three years. Woodland refers to the forest land where trees, shrubs, bamboos and coastal mangroves grow. Grassland refers to all kinds of grassland mainly growing herbaceous plants with coverage of more than 5%, including shrub grassland mainly for grazing and sparse forest grassland with canopy density of less than 10%. Built-up land refers to urban and rural residential areas and industrial, mining, transportation and other land out-side. Unused land refers to the land that has not been used at present, including the land that is difficult to use. Including sandy land, gobi, saline alkali land, marsh land, bare land, bare rocky land, alpine desert, tundra, etc.

Commented 25: The scale does not look right.

Response: Thank you for your opinions. We have modified the scale of Figure 4 and redrawn it. Please see the Figure 4, as follows:

Figure 4. Spatial change of land use in the YRB between 2000 and 2020. (1) and (2) represent two areas of significant change randomly selected.

Commented 26: The statement “In 2020, soil erosion in cropland and woodland continued to decline” means that SE from January to December 2020 constantly decreased. However there are no such data in Fig. 5. There are only 2000, 2010 & 2020. But, from 2010 to 2020, they were not decreasing, BUT slightly increasing???

Response: Thank you for your comments. We have revised“In 2020, the soil erosion of cropland and woodland were further decline”. We also changed the figure 5 and added digits to support it. Please see the content in lines 408-409, as follows:

In 2020, the soil erosion of cropland and woodland were further decline.

Figure 5. Soil erosion acreage and numerical value of various land usage types between 2000 and 2020

Commented 27: The use of line graph connecting SE between land-use types in Fig. 5 is not statistically correct because there is no continuity between those types of land uses. The lower right graph in Fig. 5 cannot also be figured out because of the too high & too low values plotted in one graph. It has to be presented in the form of numeric table.

Response: Thank you for your comments. We have added new Table 6. Please see the content in lines 418-419, as follows:

Table 6. Proportion of different land use areas under different soil erosion grades in the YRB from 2000 to 2020

Classification

of SE

LUCC

Proportion of soil erosion area(%)

2000

2020

Very slight

Cropland

89.53

95.38

Woodland

92.06

96.10

Grassland

77.62

81.66

Unused land

82.92

82.97

Slight

Cropland

8.48

3.75

Woodland

5.85

2.76

Grassland

14.98

12.83

Unused land

10.38

10.40

Moderate

Cropland

1.35

0.54

Woodland

1.33

0.58

Grassland

4.22

2.71

Unused land

3.21

3.00

Strong

Cropland

0.41

0.15

Woodland

0.36

0.22

Grassland

1.62

1.16

Unused land

1.51

1.55

Very strong

Cropland

0.10

0.05

Woodland

0.16

0.17

Grassland

1.00

0.93

Unused land

1.20

1.25

Severe

Cropland

0.01

0.01

Woodland

0.06

0.06

Grassland

0.48

0.60

Unused land

0.69

0.62

Commented 28: Again, please define exactly what is arable land & unused land, so that the authors & the readers will share the same perception.

Response: Thank you for your comments. We have redefined exactly what is arable land & unused land in Table 5, and we have replaced arable land with cropland arable land in whole text. Please see the content in Table 5, as follows:

Table 5. Land usage matrix for the Yellow River Basin 2000–2020 (km2)

LUCC

Cropland

Woodland

Grassland

Water area

Built-up land

Unused land

Transfer out

Total area in 2020

Cropland

18,040

75,381

5,879

35,688

3,472

138,460

349,076

Woodland

16,329

52,217

973

1,974

3,090

74,583

197,962

Grassland

72,178

56,200

12,873

8,216

87,530

236,997

841,934

Water area

4,815

803

10,387

1,242

5,525

22,772

53,999

Built-up land

21,537

788

3,639

1,945

509

28,418

61,650

Unused land

5,256

3,482

109,565

10,271

1,985

130,559

525,281

Transfer in

120,115

79,313

251,189

31,941

49,105

100,126

Total area in 2000

367,435

193,489

827,843

44,699

40,903

555,023

Note: Cropland refers to the land where crops are planted, including mature cropland, newly developed wasteland, leisure land, alternate land and grassland crop rotation land; Land for agricultural fruit, mulberry, and forestry mainly for planting crops; Beach and seashore cultivated for more than three years. Woodland refers to the forest land where trees, shrubs, bamboos and coastal mangroves grow. Grassland refers to all kinds of grassland mainly growing herbaceous plants with coverage of more than 5%, including shrub grassland mainly for grazing and sparse forest grassland with canopy density of less than 10%. Built-up land refers to urban and rural residential areas and industrial, mining, transportation and other land out-side. Unused land refers to the land that has not been used at present, including the land that is difficult to use. Including sandy land, gobi, saline alkali land, marsh land, bare land, bare rocky land, alpine desert, tundra, etc.

Commented 29: The scale in this does not look right.

Response: Thank you for your opinions. We have modified the scale of Figure 7 and redrawn it. Please see the Figure 7, as follows:

Figure 7. Maps of SE and land usage under natural development and ecological protection in 2040. (1), (2), (3) and (4) represents areas of significant change randomly selected.

Commented 30: By using a GIS, the total area for each type of land use can be extracted from the map, and in order not to confuse the readers, these extracted data should be presented in the form of Table. The scale of the map also does not look right? Again (1) & (2) are different positions in Figure 7, please explain in

the text what are they; why their spatial positions are different between maps?

Response: Thank you for your opinions. We have used GIS to extract statistics on land use types and soil erosion, and the specific results are shown in Table 7. Because the study area is large, the difference between land use maps and soil erosion maps in different years seems not obvious. In order to make readers see the changes in different years more clearly, different regions with obvious changes are selected as enlarged maps. Therefore, positions (1), (2), (3)and(4) of Figure 7 are different. We have explained the different locations. Please see the Table 7 and Figure 7, as follows:

Table 7. Proportion of land use area and soil erosion characteristics under different scenarios in the YRB in 2040

LUCC

Natural development scenario

Ecological protection scenario

Cropland

17.07%

17.04%

Woodland

10.05%

10.14%

Grassland

41.74%

41.85%

Water area

2.67%

2.67%

Built-up land

3.42%

3.38%

Unused land

25.06%

24.91%

Soil erosion

4.81 t hm−2 a−1

4.78 t hm−2 a−1

Figure 7. Maps of SE and land usage under natural development and ecological protection in 2040. (1), (2), (3) and (4) represents areas of significant change randomly selected.

Commented 31: Where this data from? Were there some ground measurements?

Response: Thank you for your comments. These data are based on the statistics of soil erosion, and Table 2 has been added. Please see the content in lines 471-472, as follows:

In this study, we evaluated the SE in the Yellow River Basin between 2000 and 2020 and found that the SE classification were all very slight erosion, and the actual erosion amounts were 10.13×108 t, 9.20×108 t and 8.92×108 t (Table 2).

Table 2. Characteristics of different soil erosion intensities from 2000–2020

Year

Soil erosiont hm−2 a−1

Total soil erosion (108 t)

2000

5.28

10.13

2010

4.80

9.20

2020

4.65

8.92

Commented 32: How can soil erosion consistent with Government???

Response: Thank you for your opinions. The word "consistent" is inaccurate. We have replaced it with " closely related". The Chinese government has promulgated the policy of returning farmland to forests and grasslands since 1999. These policies are beneficial to reducing soil erosion, and there are also quantitative studies on relevant literature. Please see the content in lines 485-487, as follows:

Soil erosion decreased by 11.92% from 2000 to 2020, which is closely related with the program of converting farmland to grassland, which was conducted by the Chinese government in 1999.

Commented 33: This is not consistent with the data because in the explanation of Fig.7, the area of unused land was much higher than arable land area.

Response: Thank you for your comments. We have modified according to Table 7, and the unused land is higher than the cropland. Please see the content in line 479, as follows:

The main land types are grassland, unused land, cropland, and woodland in the YRB,

Table 7. Proportion of land use area and soil erosion characteristics under different scenarios in the YRB in 2040

LUCC

Natural development scenario

Ecological protection scenario

Cropland

17.07%

17.04%

Woodland

10.05%

10.14%

Grassland

41.74%

41.85%

Water area

2.67%

2.67%

Built-up land

3.42%

3.38%

Unused land

25.06%

24.91%

Soil erosion

4.81 t hm−2 a−1

4.78 t hm−2 a−1

Commented 34: What types of their characteristics were decreasing and increasing?

Response: Thank you for your comments. The cropland and unused land areas showing decreasing trend and woodland and grassland showing increasing trend. Please see the content in lines 480-482, as follows:

and the transformation between the four is clearly defined, with cropland and unused land areas showing decreasing trend and woodland and grassland showing increasing trend.

Commented 35: What are the relationships between those projects and the erosion characteristics of the area so that they need to be included in the discussion? This should be discussed further.

Response: Thank you for your opinions. These projects implemented by the Chinese government in 1999 have changed the surface environment, and the area of forest land and grassland is increasing. These measures have effectively reduced water and soil loss. Relevant literature has been studied quantitatively. Therefore, we want to find out the reasons for the reduction of water and soil loss from the perspective of policy and discuss them. Please see the content in lines 482-490, as follows:

China invested a total of US$378.5 billion in sustainable development projects targeting land systems in 1978-2015 [63] and forest ecological conservation. Grassland sys-tem projects such as restoration of forest ecology, grassland systems, soil erosion reduced by 12.9% in China from 2000 to 2010 [64]. In the Loess Plateau, large-scale restoration and afforestation of cropland and barren land has reduced soil erosion to historically low levels [65]. and agro-industrial development have considerably im-proved the natural environment and quality of life in rural areas. The supplementary consolidation of valley bottoms cropland in China alleviated the pressure of agricultural development on sloping land and reduced soil erosion by a further 10% [66].

Commented 36: A scientific discussion should be based on relevant data, not just saying … considerably improved … etc without presenting the evidence for that.

Response: Thank you for your comments. We have supplemented relevant references and provided relevant data for support. Please see the content in lines 484-490, as follows:

Grassland system projects such as restoration of forest ecology, grassland systems, soil erosion reduced by 12.9% in China from 2000 to 2010 [64]. In the Loess Plateau, large-scale restoration and afforestation of cropland and barren land has reduced soil erosion to historically low levels [65]. and agro-industrial development have considerably im-proved the natural environment and quality of life in rural areas. The supplementary consolidation of valley bottoms cropland in China alleviated the pressure of agricultural development on sloping land and reduced soil erosion by a further 10% [66].

  1. Ouyang, Z.Y.; Zheng, H.; Xiao, Y.; Polasky, S.; Liu, J.G.; Xu, W.H.; Wang, Q.; Zhang, L.; Xiao, Y.; Rao, E.M.; Jiang,L.; Lu,F.; Wang,X.K.; Yang,G.B.; Gong, S.H.; Wu, B.F.; Zeng,Y.; Yang, W.; Daily, G. C. Improvements in ecosystem services from in-vestments in natural capital. Science 2016, 352, 1455–1459.
  2. Yang, H. F., Mu, S. J. & Li, J. L. Efects of ecological restoration projects on land use and land cover change and its infuences on territorial NPP in Xinjiang, China. Catena 2014, 115, 85–95.
  3. Liu, Y. S., Guo, Y. J., Li, Y. R. & Li, Y. H. GIS-based efect assessment of soil erosion before and after gully land consolidation: a case study of Wangjiagou project region, Loess Plateau. Chin. Geogr. Sci. 2015, 25, 137–146.

Commented 37: Where is the evidence of relationship? Was reduction in cropland area related to reduced erosion in the cropland or the total erosion in the study area? What are the characteristics of the cropland? Surface erosion is more dependent on the slope and rainfall intensity and magnitude, isn’t it?

Response: Thank you for your opinions. We have extracted the specific change rate from the result analysis, and the reduction of cropland area has a certain relationship with soil erosion (Figure 6). Although surface erosion is more dependent on the slope and rainfall intensity and magnitude, but some cropland is planted on steep slopes, These slope cropland also suffer from soil erosion during rainfall. Some studies have found that the supplementary consolidation of valley bottoms cropland in China alleviated the pressure of agricultural development on sloping land and reduced soil erosion by a further 10%. [Liu, Y. S., Guo, Y. J., Li, Y. R. & Li, Y. H. GIS-based efect assessment of soil erosion before and after gully land consolidation: a case study of Wangjiagou project region, Loess Plateau. Chin. Geogr. Sci. 25, 137–146 (2015).]. This is consistent with our research results.

Figure 6. Links between SE and land usage alteration from 2000 to 2020

Commented 38: So, these results need to be presented in the forms of easy to understand with well defined terms such as unused land, arable land, transfer-in, transfer-out, etc, and the terms used are the same between in the tables and in the narrations in the text.

Response: Thank you for your opinions. We have added Table 2 and Table 7, and these results are easier to understand in the table. We have also defined the terms in the table and uniformly revised them in the full text.

Commented 39: The main cause of SE is the intensity and frequency of rainfall and slope characteristics of land area. For example, monthly rainfall of 300 mm with 30 rainy days may result in zero SE, but if the 300 mm rain in one or two consecutive days, it will definitely cause significant SE on steep lands even if it is a grassland.

Response: Thank you for your opinions. We agree with you that the intensity and frequency of rainfall and the slope characteristics of the land area have a significant impact on soil erosion. However, land use is greatly affected by human activities and is relatively intuitive. Rainfall changes are difficult to predict, and the slope changes are relatively small. Therefore, this study aims to find out the relationship between land use and soil erosion by predicting land use changes, It provides basis for rational use of land and soil erosion control.

Commented 40: These have to be clearly stated in the abstract (which are not appear) so that the readers & policy makers will understand what this paper is trying to present. In addition, GIS modeling has to incorporate ground measurement for model calibration and to bring modeling results to as close as possible to the real conditions on the ground.

Response: Thank you for your comments. We have made changes in the summary. Added:) The intensity and frequency of rainfall and the slope characteristics of the land area have a significant impact on soil erosion. However, land use is greatly affected by human activities and is relatively intuitive. Rainfall change is difficult to predict, while the slope change is relatively small. Therefore, the RUSLE factor sub P is associated with LUCC data in 2040 to predict and assess the soil erosion of the YRB in 2040 under natural development and ecological protection scenarios. Please see the content in lines 15-40, as follows:

Abstract: Soil erosion is an important global environmental issue that severely affects the regional ecological environment and socio-economic development. The Yellow River (YR) is China’s second largest river and the fifth largest one worldwide. Its watershed is key to China's economic growth and environmental security. In this study, six impact factors including: rainfall erosivity (R), soil erosivity (K), slope length (L), slope steepness (S), cover management (C) and protective measures (P) were used. Based on the revised universal soil loss formula (RUSLE) mode, and combined with geographic information system (GIS) to estimate temporal and spatial distribution of soil erosion (SE) in the YR from 2000 to 2020. The patch-generating land use simulation (PLUS) model was used to simulate the land use and land cover change (LUCC) under two scenarios (natural development and ecological protection in 2040, and the RUSLE factor P is associated with LUCC in 2040, and the soil erosion in the Yellow River Basin (YRB) in 2040 under two scenarios were predicted and evaluated. This method has great advantages in land use simulation, but soil erosion is greatly affected by rainfall and slope and focuses on the link between land-usage alteration and SE. Therefore, this method has certain limitations in assessing soil erosion by simulating and predicting land use change. We found that there is generally slight soil erosivity in the YRB, with the most serious soil erosion occurring in 2000. Areas for serious SE are predominantly situated occur in the upper reaches (URs), followed by the middle reaches (MRs), and is less severe in the lower reaches. Soil erosion in the YRB has decreased 11.92% from 2000 to 2020, and the soil erosion has gradually reduced. Land-use change strongly influences SE while an increase of the woodland area has an important positive effect in reducing soil erosion. By predicting land use changes in 2040, compared with the natural scenario, woodland and grassland under the ecological conservation scenario will be increase by 1,978 km2 and 2,407 km2, respectively. Soil erosion will be decrease by 6.24%, and the implementation of woodland and grassland protection helped reduce soil erosion. Policies such as forest protection and grassland restoration should be further developed and implemented on the MRs and URs of the YR. Our research results possess important trend-setting significance for soil erosion control protocols and ecological environmental protection in other large river basins worldwide.

Commented 41: This must be clearly stated in the abstract and incorporated in formulating the objective of this study, and this was not clear in the abstract.

Response: Thank you for your opinions. We have clearly stated in the abstract and incorporated in formulating the objective of this study. Please see the content in lines 15-40, 120-128, as follows:

Abstract: Soil erosion is an important global environmental issue that severely affects the regional ecological environment and socio-economic development. The Yellow River (YR) is China’s second largest river and the fifth largest one worldwide. Its watershed is key to China's economic growth and environmental security. In this study, six impact factors including: rainfall erosivity (R), soil erosivity (K), slope length (L), slope steepness (S), cover management (C) and protective measures (P) were used. Based on the revised universal soil loss formula (RUSLE) mode, and combined with geographic information system (GIS) to estimate temporal and spatial distribution of soil erosion (SE) in the YR from 2000 to 2020. The patch-generating land use simulation (PLUS) model was used to simulate the land use and land cover change (LUCC) under two scenarios (natural development and ecological protection in 2040, and the RUSLE factor P is associated with LUCC in 2040, and the soil erosion in the Yellow River Basin (YRB) in 2040 under two scenarios were predicted and evaluated. This method has great advantages in land use simulation, but soil erosion is greatly affected by rainfall and slope and focuses on the link between land-usage alteration and SE. Therefore, this method has certain limitations in assessing soil erosion by simulating and predicting land use change. We found that there is generally slight soil erosivity in the YRB, with the most serious soil erosion occurring in 2000. Areas for serious SE are predominantly situated occur in the upper reaches (URs), followed by the middle reaches (MRs), and is less severe in the lower reaches. Soil erosion in the YRB has decreased 11.92% from 2000 to 2020, and the soil erosion has gradually reduced. Land-use change strongly influences SE while an increase of the woodland area has an important positive effect in reducing soil erosion. By predicting land use changes in 2040, compared with the natural scenario, woodland and grassland under the ecological conservation scenario will be increase by 1,978 km2 and 2,407 km2, respectively. Soil erosion will be decrease by 6.24%, and the implementation of woodland and grassland protection helped reduce soil erosion. Policies such as forest protection and grassland restoration should be further developed and implemented on the MRs and URs of the YR. Our research results possess important trend-setting significance for soil erosion control protocols and ecological environmental protection in other large river basins worldwide.

Our specific objectives are: (1) to reveal the changes of soil erosion in the YRB from 2000 to 2020, and the characteristics of changes under different scenarios; (2) to relationship between land use change and soil erosion was discussed, and the impact of land use change on soil erosion was analyzed quantitatively; (3) based on PLUS model, the land use change is predicted, and the soil erosion in the YRB in 2040 is predicted in combination with the P-factor of RUSLE model. It reveals the changes in SE under different scenarios, supplies a scientific basis for more science driven erosion control, ecological zoning, and coordinated improvement of the economic environment in the YRB.

Commented 42: These have to be clearly stated in the Abstract.

Response: Thank you for your comments. We have clearly stated in the abstract. Please see the content in lines 26-29, as follows:

This method has great advantages in land use simulation, but soil erosion is greatly affected by rainfall and slope and focuses on the link between land-usage alteration and SE. Therefore, this method has certain limitations in assessing soil erosion by simulating and predicting land use change.

Commented 43: Was this relationship only in modeling calculation or based on ground measurement or both. Please clearly defined in the method and in the abstract.

Response: Thank you for your opinions. This relationship only in modeling calculation. Please see the content in lines 33-34,499-502, as follows:

Based on GIS statistics, land-use change strongly influences SE while an increase of the woodland area has an important positive effect in reducing soil erosion.

In 20 years, the cropland has decreased by 5.0%, the soil erosion has decreased by 55.46%, the forestland has increased by 2.31%, the soil erosion has decreased by 42.34%, the grassland has increased by 1.70%, and the soil erosion has decreased by 13.36% in YRB (Figure 6).

Commented 44: This was also not clearly defined, and please do so. Even on a neglected land, shrub and grass can grow well.

Response: Thank you for your comments. We have defined the unused land. As the area of unused land in the study area is relatively large, the analysis of the results of this study shows that we should increase the development efforts and change to forestland and grassland. Please see the content in lines 358-361, as follows:

Unused land refers to the land that has not been used at present, including the land that is difficult to use; Including sandy land, gobi, saline alkali land, marsh land, bare land, bare rocky land, alpine desert, tundra, etc.

Reviewer 2 Report

Dear Authors

The article concerns the temporal and spatial variability of soil erosion in the years 2000-2020 in the Yellow River basin. Two models, RUSLE and PULS, were used in the study. The work is well conducted and informative. The methodology is sound, and the results seem to be reliable. However, despite the interesting results, the article has several flaws:

1 - In section 2.2. Data sources the description of the materials used is too general. Were satellite images used in the study? or other cartographic materials? The Internet addresses provided in most cases refer to Chinese Internet portals, which are practically useless for people without knowledge of the Chinese language.

2 - The article lacks information about the importance of the conducted research for public health?

3 - The authors in the article often use the names of Chinese provinces. For people outside of China, the location of individual provinces in the Yellow River catchment area is very problematic. Maybe it's worth to include in the article a drawing with the location of the more important provinces that are mentioned in the manuscript?

4 - In Fig. 2, there is no Yellow River in the diagram with DEM, the color scale is too small, it should be more extended, it is worth specifying some intermediate values, not just the min-max range.

5 - Where are the soil erosion values given on line 248: in 2000, 2010, and 2020 is 5.28, 4.8 and 4.65 t hm−2 a−1. There are no such parameters in Table 2

6 - In table 3, the total area in 2000 is 2,029,902 and in 2020 - 2,029,392, where does this difference come from?

7 - Why in figures 3, 4 and 5 different areas (1) and (2) are shown each time. What made you choose these areas?

8 - The first paragraph in the conclusions section is not a conclusion but contains the location of the study area and a general description of the method used. The conclusions section needs rework. Conclusions in their current form can be formulated without any research.

9 - No reference in the text of Fig. 1.

10 - On line 353 there is a reference to table 5 in the article there is no such table. This is table number 4.

11 - No reference in the text of items 38 and 42 from the references.

Author Response

Point–to–point responses

Reviewer #2

Commented 1: In section 2.2. Data sources the description of the materials used is too general. Were satellite images used in the study? or other cartographic materials? The Internet addresses provided in most cases refer to Chinese Internet portals, which are practically useless for people without knowledge of the Chinese language.

Response: Thanks for your comments. We discussed the data sources in detail. This research mainly uses the satellite image data of the Chinese Academy of Sciences and the vector data of administrative divisions. We have supplemented the DEM and soil data source websites for English browsing and data acquisition. Please see the content in lines 153-165, as follows:

2.2 Data sources

This study mainly uses vector data, DEM data, land use data, meteorological data, NDVI data, soil data, road network data and socio-economic data. Among them, vector data refers to the boundary contour of the study area and the location of China's boundary contour. According to the “China Land Use/Land Cover Remote Sensing Monitoring Data Classification System” the land use data were reclassified into six categories: cropland, forestland, grassland, water area, built-up land and unused land. Soil data is the extraction of clay, silt, sand and organic carbon. Meteorological data refers to the temperature and precipitation data in the study area, which are obtained by interpolation using Anusplin software according to the meteorological station data. Road network data mainly refers to high-speed railway, main trunk roads and secondary roads. Datasets were obtained online with a resolution of 1 × 1 km, and for maintaining data resolution consistency, data higher than 1 km resolution were uniformly resampled using ArcGIS 10.2 software. Data was collected for 2000–2020, and the specific data sources are exhibited in Table 1.

Table 1. List of data sets and their original sources

Data type

Data source

Resolution ratio

Time

DEM

https://earthexplorer.usgs.gov/

30 m

LUCC

https://www.resdc.cn

1 km

2000–2020

Soil

http://www.fao.org/land water/databases-and-software/hwsd/en/

1 km

Meteorological

https://data.cma.cn/

1 km

2000–2020

NDVI

https://modis.gsfc.nasa.gov/

250 m

2000–2020

Road

https://www.openstreetmap.org/

1 km

2020

POP

https://www.resdc.cn/

1 km

2019

GDP

https://www.resdc.cn/

1 km

2019

Commented 2: The article lacks information about the importance of the conducted research for public health?

Response: Thanks for your comments. Soil erosion will not only cause land degradation, reduce land productivity and affect agricultural production, but also cause rivers, lakes and reservoirs to be blocked, water quality to be polluted, drought and waterlogging disasters to be aggravated, which seriously threatens human public health and safety. We have added a description of public health in the introduction. Please see the content in lines 45-48, as follows:

Soil erosion (SE) severely affects normal development of agriculture and industry, decreases soil fertility, leads to ecological degradation, and damages water and transportation engineering facilities. It is a global issue that has led to land degrada-tion and impairment of ecosystem services and serious threat to human public health safety [1–2].

Commented 3: The authors in the article often use the names of Chinese provinces. For people outside of China, the location of individual provinces in the Yellow River catchment area is very problematic. Maybe it's worth to include in the article a drawing with the location of the more important provinces that are mentioned in the manuscript?

Response: Thanks for your comments. We have redrawn the location Figure 2 of the study area, adding the names of nine provinces and 73 cities in the Yellow River Basin to facilitate readers' understanding of the location of each administrative region. Please see the Figure 2, as follows:

Figure 2. Elevation map of the investigation region

Commented 4: In Fig. 2, there is no Yellow River in the diagram with DEM, the color scale is too small, it should be more extended, it is worth specifying some intermediate values, not just the min-max range.

Response: Thanks for your suggestions. We have added the Yellow River to the DEM, and the DEM values are divided into 6 levels. Please see the Figure 2, as follows:

Figure 2. Elevation map of the investigation region

Commented 5: Where are the soil erosion values given on line 248: in 2000, 2010, and 2020 is 5.28, 4.8 and 4.65 t hm−2 a−1. There are no such parameters in Table 2

Response: Thanks for your comments. We have added new Table 2 and relevant values. Please see the content in lines 298-300, as follows:

By calculating the soil erosion of the YRB from 2000 to 2020 and calculating the specific amount of soil erosion, the results of SE changes in the study area are shown in Table 2.

Table 2. Characteristics of different soil erosion intensities from 2000 to 2020

Year

Soil erosiont hm−2 a−1

Total soil erosion (108 t)

2000

5.28

10.13

2010

4.80

9.20

2020

4.65

8.92

Commented 6: In table 3, the total area in 2000 is 2,029,902 and in 2020 - 2,029,392, where does this difference come from?

Response: Thanks for your suggestions. We recalculated the total area in Table 5, which he total area in 2000 is 2,029,902 km2 and in 2020 - 2,029,392 km2. The main reason for this difference is that the resolution of our land use data is 1×1 km, the resolution is low, and the area of the study area is large, so the land use area of the two phases is different. We also compared the relevant literature, and found this problem, but the gap is not large, which is acceptable. Reference as follows:

Gilani H, Ahmad A, Younes I, Abbas S. Impact assessment of land cover and land use changes on soil erosion changes (2005–2015) in Pakistan. Land Degrad Dev.2022, 33:204–217.

Commented 7: Why in figures 3, 4 and 5 different areas (1) and (2) are shown each time. What made you choose these areas?

Response: Thanks for your comments. Because the study area is large, the difference between land use maps and soil erosion maps in different years seems not obvious. In order to make readers see the changes in different years more clearly, different regions with obvious changes are selected as enlarged maps. Therefore, positions (1) and (2) of Figures 3, 4 and 5 are different.

Commented 8: The first paragraph in the conclusions section is not a conclusion but contains the location of the study area and a general description of the method used. The conclusions section needs rework. Conclusions in their current form can be formulated without any research.

Response: Thanks for your opinions. We deleted the first paragraph of the conclusion and reorganized the whole conclusion. Please see the content in lines 533-554, as follows:

  1. Conclusions

Our research mainly discussed the relationship between land use change and soil erosion in the YRB from 2000 to 2020, revealed the impact of land use change on soil erosion, and studied the characteristics of soil erosion under different scenarios in the YRB in the future.

We found that the soil erosion in the YRB from 2000 to 2020 was slight, and the main land use types were grassland, unused land, cropland and woodland. Soil ero-sion is significantly mitigated. There is a direct relationship between cropland and soil erosion, with a decrease in the cropland area reducing soil erosion. There is an inverse relationship between woodland and grassland, where an increase in the area reduces soil erosion. Woodland had the best effect in terms of mitigating soil erosion. In 2040, compared with the natural development scenario, the ecological protection scenario in the Yellow River basin will increase the forest land and grassland by 1978km2 and 2407km2 respectively, and the soil erosion will decrease by 6.24%. Without consider-ing climate change, soil erosion pressure will increase in the future, but the ecological protection scenario can effectively reduce soil erosion. The study area has a large proportion of unused land, so it is recommended to increase the development of un-used land and transform it into forested grassland. According to the topographic con-ditions, some grasslands should be converted into forested land, increase the propor-tion of forested land, and strengthen measures, for example sloping cropland im-provement. Our conclusions provide a scientific basis for future land use planning, soil and water loss management and prevention in the Yellow River Basin.

Commented 9: No reference in the text of Fig. 1.

Response: Thanks for your opinions. We have added Figure 1 to the text. Please see the content in lines 118-120, as follows:

In the investigation, the spatiotemporal alterations in soil erosion in 73 prefec-ture-level cities in the YRB were analyzed from 2000 to 2020 using the RUSLE model to explore links between land usage change and SE (Figure 1).

Commented 10: On line 353 there is a reference to table 5 in the article there is no such table. This is table number 4

Response: Thanks for your comments. We have changed Table 5 to Table 4. Please see the content in lines 428-429, as follows:

Combined with the land-use matrix (Table 4), the transfer of woodland and grassland was greater than the transfer out.

Commented 11: No reference in the text of items 38 and 42 from the references.

Response: Thanks for your opinions. We have rearranged the references of the whole text, and without omission. There are 73 references in this paper, all of which can be found in the text.

Round 2

Reviewer 2 Report

Dear Authors

Thank you for giving me the opportunity to review the article again. My comments were taken into account by the authors. The article can be published in this version.